# A convolutional neural network highlights mutations relevant to antimicrobial resistance in *Mycobacterium tuberculosis*

Anna G. Green [1,7], Chang Ho Yoon [1,2,7], Michael L. Chen[1,3], Yasha Ektefaie[1], Mack Fina[4], Luca Freschi[1], Matthias I. Gröschel [1], Isaac Kohane[1], Andrew Beam[1,5,7 ✉] & Maha Farhat [1,6,7 ✉]

Long diagnostic wait times hinder international efforts to address antibiotic resistance in *M. tuberculosis*. Pathogen whole genome sequencing, coupled with statistical and machine learning models, offers a promising solution. However, generalizability and clinical adoption have been limited by a lack of interpretability, especially in deep learning methods. Here, we present two deep convolutional neural networks that predict antibiotic resistance phenotypes of *M. tuberculosis* isolates: a multi-drug CNN (MD-CNN), that predicts resistance to 13 antibiotics based on 18 genomic loci, with AUCs 82.6-99.5% and higher sensitivity than state-of-the-art methods; and a set of 13 single-drug CNNs (SD-CNN) with AUCs 80.1-97.1% and higher specificity than the previous state-of-the-art. Using saliency methods to evaluate the contribution of input sequence features to the SD-CNN predictions, we identify 18 sites in the genome not previously associated with resistance. The CNN models permit functional variant discovery, biologically meaningful interpretation, and clinical applicability.

[1] Department of Biomedical Informatics, Harvard Medical School, 25 Shattuck St, Boston, MA 02115, USA. [2] Big Data Institute, Nuffield Department of Population Health, University of Oxford, Oxford OX37LF, UK. [3] Stanford University School of Medicine, 291 Campus Dr, Stanford, CA 94305, USA. [4] Harvard College, Cambridge, MA 02138, USA. [5] Department of Epidemiology, Harvard T.H. Chan School of Public Health, 677 Huntington Ave, Boston, MA 02115, USA. [6] Division of Pulmonary & Critical Care, Massachusetts General Hospital, 55 Fruit St, Boston, MA 02114, USA. [7] These authors contributed equally: Anna G. Green, Chang Ho Yoon, Andrew Beam, Maha Farhat. ✉email: Andrew_Beam@hms.harvard.edu; Maha_Farhat@hms.harvard.edu

Tuberculosis is a leading cause of death worldwide from an infectious pathogen, with more than 1.5 million people succumbing to the disease annually[1]. Rates of antibiotic-resistant *Mycobacterium tuberculosis*, the causative agent of tuberculosis, continue to rise, posing a threat to public health[2]. A major challenge in combatting antibiotic-resistant tuberculosis is the timely selection of appropriate treatments for each patient, particularly when growth-based drug susceptibility testing takes weeks[1].

Molecular diagnostic tests for *M. tuberculosis* antimicrobial resistance reduce diagnostic wait times to hours or days, but only target a small number of loci relevant to a few antibiotics, and cannot detect most rare genetic variants[3]. Although whole genome sequencing-related diagnostic tests offer the promise of testing many loci and inferring resistance to any drug, statistical association techniques have seen limited success, hindered by their inability to assess newly observed variants and epistatic effects[3–7]. More complex models such as deep learning provide promising flexibility but are often uninterpretable, making them difficult to audit for safety purposes[8,9]. Moreover, interrogating black box models offers the opportunity for hypothesis generation which can be later validated, potentially improving scientific understanding of the underlying phenomenon[10]. An ideal sequencing-based diagnostic method would predict resistance to any drug based on the entire genome, and rapidly provide interpretable outputs about which loci contributed to resistance predictions, allowing for such a method to greatly augment current molecular diagnostics with expanded catalogs of resistance-conferring loci, or supersede those diagnostics entirely.

A recent "wide-and-deep" neural network applied to *M. tuberculosis* genomic data outperformed previous methods to predict antimicrobial resistance to 10 antibiotics[11]; however, like most deep learning methods, the logic behind its predictions was indiscernible. Although more interpretable, rule-based classifiers of antimicrobial resistance in *M. tuberculosis* have been developed[12,13], these rely on predetermined, single-nucleotide polymorphisms or *k*-mers, hindering their flexibility to generalize to newly observed mutations, and universally ignore genomic context. Deep convolutional neural networks (CNNs), which greatly reduce the number of required parameters compared to traditional neural networks, could be used to consider multiple complete genomic loci with the ultimate goal of incorporating the whole genome. This would allow the model to assess mutations in their genetic context by capturing the order and distance between resistance mutations of the same locus, allowing a better incorporation of rare or newly observed variations. Deep CNNs, when paired with attribution methods that highlight the most salient features informing the model predictions, are a promising means of harnessing the predictive power of deep neural networks in genomics for biological discovery and interpretation[14]. The extent to which we may trust these highlighted features remains the subject of ongoing scientific exploration[8,15,16]. CNNs also have the added advantage of minimizing the preprocessing needed of genomic variant data.

Here, we show that CNNs perform on a par with the state-of-the-art in predicting antimicrobial resistance in *M. tuberculosis* and provide biological interpretability through saliency mapping. We train two models: one designed for accuracy that incorporates genetic and phenotypic information about all drugs; and a second designed for interpretability that forces the model to only consider putatively causal regions for a particular drug. Our models are trained on the entire genetic sequence of 18 regions of the genome known or predicted to influence antibiotic resistance, using data collected from over 20,000 *M. tuberculosis* strains spanning the four major global lineages. Across each locus, we calculate genomic positions that most influence the prediction of resistance for each drug, validating our method by recapitulating known positions and providing predictions of new positions potentially involved in drug resistance. Given the growing movement towards greater interpretability in machine learning methods[16,17], we expect this model to have implications for hypothesis generation about molecular mechanisms of antimicrobial resistance through genotype–phenotype association.

## Results

**Training dataset characteristics.** We train and cross-validate our models using 10,201 *M. tuberculosis* isolates from the ReSeqTB and the WHO Supranational Reference Laboratory Network (sources detailed in the "Methods" section). Each isolate is phenotyped for resistance to at least one of 13 antitubercular drugs: the four first-line drugs isoniazid, rifampicin, ethambutol, and pyrazinamide, and nine additional second-line drugs (Tables 1 and 2). All drugs are represented by at least 250 phenotyped isolates.

### Table 1 Training set isolate phenotypes.

| Drug | Resistant (*n*) | Susceptible (*n*) | Total (*n*) | Resistant proportion |
|---|---|---|---|---|
| Isoniazid | 4232 | 5723 | 9955 | 0.425 |
| Rifampicin | 3472 | 6428 | 9900 | 0.351 |
| Ethambutol | 2273 | 6390 | 8663 | 0.262 |
| Pyrazinamide | 1505 | 5393 | 6898 | 0.218 |
| Streptomycin | 2643 | 4362 | 7005 | 0.377 |
| Amikacin | 773 | 2632 | 3405 | 0.227 |
| Capreomycin | 737 | 2838 | 3575 | 0.206 |
| Kanamycin | 796 | 2502 | 3298 | 0.241 |
| Ciprofloxacin | 118 | 388 | 506 | 0.233 |
| Ofloxacin | 912 | 2246 | 3158 | 0.289 |
| Moxifloxacin | 398 | 1941 | 2339 | 0.170 |
| Levofloxacin | 66 | 189 | 255 | 0.259 |
| Ethionamide | 791 | 1647 | 2438 | 0.324 |
| Total isolates | | | 10,201 | |

Phenotypic summary of the 10,201 isolates used to train and cross-validate the models: the numbers of resistant isolates, susceptible isolates, the total tested (sum of the numbers of resistant and susceptible isolates), and the resistant proportion, with respect to each of the 13 anti-TB drugs.

### Table 2 Test set isolate phenotypes.

| Drug | Resistant (*n*) | Susceptible (*n*) | Total (*n*) | Resistant proportion |
|---|---|---|---|---|
| Isoniazid | 3384 | 8870 | 12,254 | 0.276 |
| Rifampicin | 3007 | 9708 | 12,715 | 0.236 |
| Ethambutol | 1498 | 7853 | 9351 | 0.160 |
| Pyrazinamide | 1211 | 7490 | 8701 | 0.139 |
| Streptomycin | 382 | 1756 | 2138 | 0.179 |
| Amikacin | 93 | 1481 | 1574 | 0.059 |
| Capreomycin | 61 | 1652 | 1713 | 0.036 |
| Kanamycin | 83 | 2202 | 2285 | 0.036 |
| Ofloxacin | 230 | 2897 | 3127 | 0.074 |
| Moxifloxacin | 103 | 2495 | 2598 | 0.040 |
| Levofloxacin | 85 | 49 | 134 | 0.634 |
| Total isolates | | | 12,848 | |

Phenotypic summary of the 12,848 isolates used to test the models: the numbers of resistant isolates, susceptible isolates, the total tested (sum of the numbers of resistant and susceptible isolates), and the resistant proportion, with respect to each of the 11 anti-TB drugs. Ciprofloxacin and ethionamide are excluded from the test dataset due to having fewer than 50 resistant isolates (0/2 resistant to ciprofloxacin; 12/25 resistant to ethionamide).

**Model design**. We build two models to predict antibiotic resistance phenotypes from genome sequences. The first is a multi-drug convolutional neural network (MD-CNN), designed to predict resistance phenotypes to all 13 drugs at once. The model inputs are the full sequences of 18 loci in the *M. tuberculosis* genome, selected based on known or putative roles in antibiotic resistance (Table 3). We choose the final MD-CNN architecture to maximize performance in cross-validation (Fig. 1, Supplementary Fig. 1). As superior performance of multi-task over single-task models has been demonstrated with convolutional neural networks in computer vision[18–20], the MD-CNN is designed to optimize performance by combining all genetic information and relating it to the full, drug-resistance profile. We compare the MD-CNN with 13 single-drug convolutional neural networks (SD-CNN), each of which has a single-task, single-label architecture, where only loci with previously known causal associations for any given drug are incorporated (Supplementary Fig. 2). Because the MD-CNN has access to all 18 loci related to any drug resistance, differences in performance may be attributable to the fact that certain resistance phenotypes share underlying genetic mechanisms, and/or to the presence of loci not causally related to but correlated with the drug resistance in question.

We benchmark both types of CNNs against an existing, state-of-the-art, multi-drug, wide-and-deep neural network (WDNN), and a logistic regression with L2 regularization, as these methods were found to perform similarly and outperform a random forest classifier[11]. The WDNN was also found to have higher sensitivity than existing catalog-based methods (Mykrobe and TB-Profiler) in a recent comparative study[21].

**Benchmarking CNN models against state-of-the-art**. We use 5-fold cross-validation to compare the performance of the four architectures (MD-CNN, SD-CNN, L2 regression, and WDNN[11]) on the training dataset ($N = 10{,}201$ isolates, Supplementary Table 1, Supplementary Data 1).

The mean MD-CNN AUC of 0.912 for second-line drugs is significantly higher than the mean 0.860 for L2 regression (Welch's t-test with Benjamini–Hochberg FDR $q < 0.05$), but the mean AUCs for first-line drugs (0.948 vs. 0.923) are not significantly different (Benjamini–Hochberg $q = 0.059$). The mean SD-CNN AUCs of 0.938 (first-line drugs) and 0.888

(second-line drugs) are not significantly different than for L2 regression (first-line $q = 0.18$, second-line $q = 0.12$). However, L2 regression demonstrates much wider confidence intervals than the CNN models (median 0.037 versus 0.010, IQR 0.035 versus 0.014), indicating a lack of reliability as the performance depends on the particulars of the cross-validation split (Fig. 2).

Against the state-of-the-art WDNN, the AUCs, sensitivities, and specificities of the MD-CNN are comparable: the MD-CNN's mean AUC is 0.948 (vs. 0.960 for the WDNN, $q = 0.13$) for first-line drugs, and 0.912 (vs. 0.924 for the WDNN, $q = 0.18$) for second-line drugs. The SD-CNN is less accurate than the WDNN for both first-line (Benjamini–Hochberg $q = 0.004$) and second-line drugs ($q = 0.004$, Supplementary Table 1, Fig. 2).

The SD-CNN (mean AUC of 0.938 for first-line drugs; mean AUC of 0.888 for second-line drugs) performs comparably to the MD-CNN for both first-line ($q = 0.18$) and second-line drugs ($q = 0.059$).

**CNN models generalize well on hold-out test data**. We test the generalizability and real-world applicability of our CNN models on a hold-out dataset of 12,848 isolates, which were curated on a rolling basis during our study (Table 2, see the "Methods" section). Rolling curation provides a more realistic test of generalizability to newly produced datasets. Due to rolling curation and source differences, the test dataset exhibits different proportions of resistance to the 13 drugs (e.g., isoniazid resistance in 28% vs. 43% in the training dataset). We assess generalizability of the models using phenotype data for 11 drugs in the hold-out test dataset, since it contains low resistance counts for ciprofloxacin and ethionamide.

We find that the MD-CNN generalizes well to never-before-seen data for first-line antibiotic resistance prediction, achieving mean AUCs of 0.965 (95% confidence interval [CI] 0.948–0.982) on both training and hold-out test sets for first-line drugs (Fig. 3). However, generalization for second-line drugs is mixed: for the drugs streptomycin, amikacin, ofloxacin, and moxifloxacin, the model generalizes well, achieving mean AUCs of 0.939 (CI 0.928–0.949) on the test data, compared with 0.939 (CI 0.929–0.949) on the training data. For the second-line drugs capreomycin, kanamycin, and levofloxacin, the model generalization is reduced, achieving mean AUCs of 0.831 (CI

**Table 3 Loci included in the MD-CNN and SD-CNN models.**

| Locus | Start | End | Drug(s) | Length (in H37Rv) |
|---|---|---|---|---|
| *acpM-kasA* | 2,517,695 | 2,519,365 | Isoniazid | 1670 |
| *gid* | 4,407,528 | 4,408,334 | Streptomycin | 806 |
| *rpsA* | 1,833,378 | 1,834,987 | Pyrazinamide | 1609 |
| *clpC* | 4,036,731 | 4,040,937 | Pyrazinamide | 4206 |
| *embCAB* | 4,239,663 | 4,249,810 | Ethambutol | 10,147 |
| *aftB-ubiA* | 4,266,953 | 4,269,833 | Ethambutol | 2880 |
| *rrs-rrl* | 1,471,576 | 1,477,013 | Streptomycin, Amikacin, Capreomycin, Kanamycin | 5437 |
| *ethAR* | 4,326,004 | 4,328,199 | Ethionamide | 2195 |
| *oxyR-ahpC* | 2,725,477 | 2,726,780 | Isoniazid | 1303 |
| *tlyA* | 1,917,755 | 1,918,746 | Capreomycin | 991 |
| *katG* | 2,153,235 | 2,156,706 | Isoniazid | 3471 |
| *rpsL* | 781,311 | 781,934 | Streptomycin | 623 |
| *rpoBC* | 759,609 | 767,320 | Rifampicin | 7711 |
| *fabG1-inhA* | 1,672,457 | 1,675,011 | Isoniazid, Ethionamide | 2554 |
| *eis* | 2,713,783 | 2,716,314 | Kanamycin, Amikacin | 2531 |
| *gyrBA* | 4997 | 9818 | Ciprofloxacin, Levofloxacin, Moxifloxacin, Ofloxacin | 4821 |
| *panD* | 4,043,041 | 4,045,210 | Pyrazinamide | 2169 |
| *pncA* | 2,287,883 | 2,289,599 | Pyrazinamide | 1716 |

The 18 loci included in the MD-CNN and their start and end coordinates (in H37Rv numbering). Each locus is designated as putatively involved in resistance to at least one drug. To construct the 13 SD-CNN models, the relevant loci for each drug are combined—for example, the isoniazid (INH) model contains the *acpM-kasA, oxyR-ahpC, katG*, and *fabG1-inhA* loci.

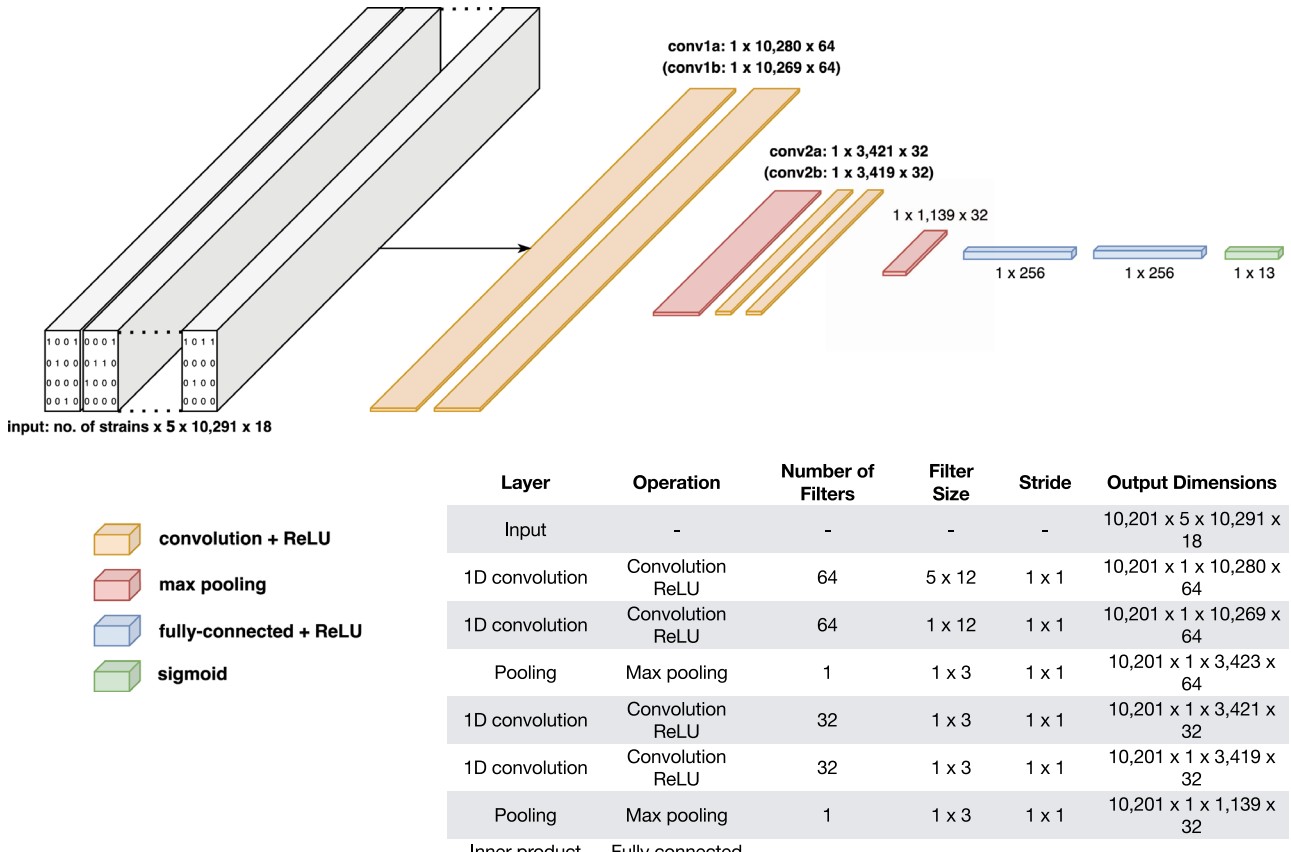

| Layer | Operation | Number of Filters | Filter Size | Stride | Output Dimensions |
|---|---|---|---|---|---|
| Input | - | - | - | - | 10,201 x 5 x 10,291 x 18 |
| 1D convolution | Convolution ReLU | 64 | 5 x 12 | 1 x 1 | 10,201 x 1 x 10,280 x 64 |
| 1D convolution | Convolution ReLU | 64 | 1 x 12 | 1 x 1 | 10,201 x 1 x 10,269 x 64 |
| Pooling | Max pooling | 1 | 1 x 3 | 1 x 1 | 10,201 x 1 x 3,423 x 64 |
| 1D convolution | Convolution ReLU | 32 | 1 x 3 | 1 x 1 | 10,201 x 1 x 3,421 x 32 |
| 1D convolution | Convolution ReLU | 32 | 1 x 3 | 1 x 1 | 10,201 x 1 x 3,419 x 32 |
| Pooling | Max pooling | 1 | 1 x 3 | 1 x 1 | 10,201 x 1 x 1,139 x 32 |
| Inner product (two times) | Fully connected ReLU | - | - | - | 256 |
| Output | - | - | - | - | 13 |

**Fig. 1 Schematic diagram and table of the multi-drug convolutional neural network (MD-CNN).** In the output layer, each of the 13 nodes is composed of a sigmoid function to compute a probability of resistance for their respective anti-TB drug (13 anti-TB drugs in total). The input consists of 10,201 isolates (TB strains) for which there is resistance phenotype data for at least 1 anti-TB drug; 5 for one-hot encoding of each nucleotide (5 dimensions, one for each nucleotide—adenine, thymine, guanine, cytosine -- plus a gap character); 10,291 being the number of nucleotides of the longest locus (*embC-embA-embB*); 18 loci of interest are incorporated as detailed in the "Methods" section.

0.824–0.838) on the test data, compared with 0.955 (CI 0.931–0.978) on the training data. We find that the SD-CNN generalizes well on first-line drug resistance for hold-out test data, with a mean AUC of 0.956 (CI 0.929–0.974). The SD-CNN also generalizes well for second-line drugs, with a mean AUC of 0.862 (CI 0.830–0.894).

We test the hypothesis that missed resistance (false negatives) in the SD-CNNs is due to mutations affecting phenotype found outside of the incorporated loci. To achieve this, we compute the number of mutations in the incorporated loci that separate each test isolate from the nearest isolate(s) in the training set and the corresponding phenotype of the nearest isolates (see the "Methods" section). We find that many of the false negatives have a genomically identical yet sensitive isolate in the training set, ranging from a minimum of 34% for pyrazinamide to a maximum of 86% for kanamycin, and suggesting that additional mutations outside of the examined loci may influence the resistance phenotype. Indeed, when considering the entire genome, almost no false negative test isolates are identical to a sensitive isolate in the training set (<6% of isolates for all drugs), indicating that additional genetic variation does exist and may lead to the currently unexplained resistance.

**MD-CNN model has improved sensitivity compared to WHO catalog.** An important feature of the CNN models is the ability to tune the model threshold to optimize sensitivity or specificity, depending on the application. We choose a threshold for all of our machine learning models (MD-CNN, SD-CNN, logistic regression + L2, and WDNN) that maximizes the sum of sensitivity and specificity. We find that the MD-CNN has the highest sensitivity of the four models for first-line (mean sensitivity 91.9%) and second-line drugs (mean sensitivity 91.1%) except ethambutol, for which the WDNN exhibits the highest sensitivity (Supplementary Table 2). The SD-CNN demonstrates the greatest specificity for first-line drugs (mean specificity 94.1%) except ethambutol where the MD-CNN has the highest; the SD-CNN demonstrates the highest specificity for second-line drugs (mean specificity 94.3%) except ethionamide and ciprofloxacin where the MD-CNN is highest (Supplementary Table 2).

Using both training and hold-out test data, we then compare the sensitivity and specificity of the MD-CNN to the field-standard WHO catalog of known, resistance-conferring variants (see the "Methods" section). In general, we find higher sensitivity for the MD-CNN model versus the WHO catalog (mean sensitivity 91.9% for first-line drugs [MD-CNN] vs. 80.4% [WHO catalog]; 91.1% for second-line drugs [MD-CNN] vs. 73.1% [WHO catalog]) at the expense of lower specificity (92.3% for first-line drugs [MD-CNN] vs. 94.8% [WHO catalog]; 85.9% for second-line drugs [MD-CNN] vs. 93.6% [WHO catalog]) (Supplementary Table 3).

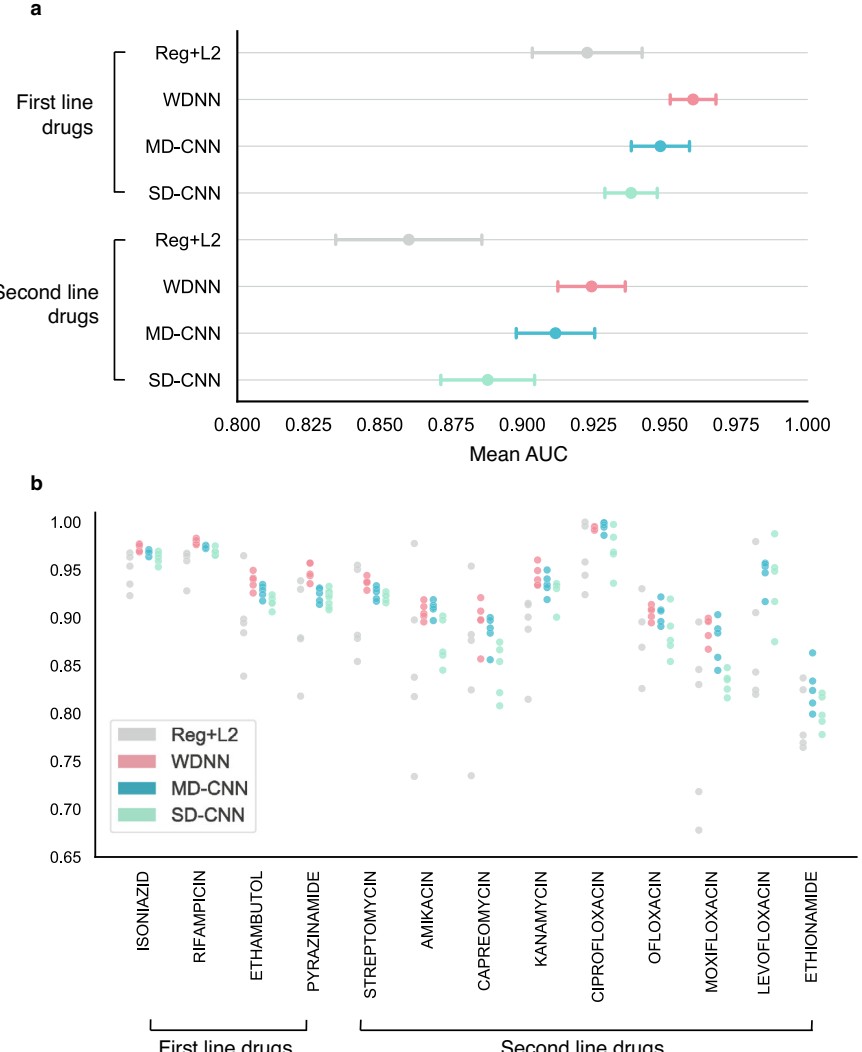

**Fig. 2 MD-CNN performs comparably to state-of-art WDNN for both first- and second-line drugs.** Results of five-fold cross validation on the training dataset ($N = 10,201$ isolates) for the four models: WDNN, logistic regression + L2 benchmark, SD-CNN, and MD-CNN. **a** Data are presented as pooled mean AUCs ± 95% confidence intervals for first-line (5 AUC values per drug, 4 drugs) and second-line drugs (5 AUC values per drug, 9 drugs). **b** data are presented as individual AUC values for each cross-validation split. The WDNN was not initially trained on levofloxacin or ethionamide and thus was not evaluated for these drugs.

**CNN models generalize on new data with realistic resistance proportions.** Datasets of drug resistance phenotypes in *M. tuberculosis* are enriched in resistant isolates compared to real global resistance frequencies. To assess the generalizability of our methods in a more "realistic" setting, we generate down-sampled, hold-out test data from the CRyPTIC database[22] with 95% pan-susceptible isolates and 5% rifampicin-resistant isolates to approximate the global prevalence[23] (see the "Methods" section). CRyPTIC provides genotype and phenotype data for the first-line drugs rifampicin, isoniazid, and ethambutol; and for the second-line drugs levofloxacin, amikacin, moxifloxacin, kanamycin, and ethionamide.

On the down-sampled CRyPTIC dataset, the MD-CNN produces mean sensitivities of 97.6% (95% confidence interval, CI, 97.3–97.8) [first-line drugs], and 86.6% (CI 85.7–87.6) [second-line drugs]; and mean specificities of 97.4% (CI 97.2–97.7) [first-line drugs], and 98.5% (CI 97.5–99.4) [second-line drugs]. For the SD-CNN, mean sensitivities are 96.2% (CI 95.9–96.5) [first-line drugs], and 87.7% (CI 86.8–88.5) [second-line drugs]; mean specificities are 97.5% (CI 97.2–97.8) [first-line drugs], and 98.9% (CI 98.1–99.7) [second-line drugs] (Supplementary Table 4).

When applied to the whole, hold-out test CRyPTIC data, the models have lower performance, particularly specificity: the MD-CNN's mean sensitivities are 96.4% (CI 96.36–96.44) [first-line drugs] and 83.3% (CI 83.2–83.5) [second-line drugs]; mean specificities are 92.7% (CI 92.66–92.75) [first-line drugs] and 92.6% (CI 92.5–92.8) [second-line drugs]. For the SD-CNN, mean sensitivities are 95.3% (CI 95.24–95.34) [first-line drugs] and 85.4% (CI 85.2–85.5) [second-line drugs]; mean specificities are 94.9% (CI 94.83–94.92) [first-line drugs] and 96.9% (CI 96.8–97.1) [second-line drugs] (Supplementary Table 4).

**MD-CNN achieves accuracy by learning dependency structure of drug resistances.** Because the inputs to the CNN models are the complete sequence of 18 genetic loci involved in drug resistance, we are able to assess the contribution of every site, in its neighboring genetic context, to the prediction of antibiotic resistance phenotypes. We do this by calculating a saliency score

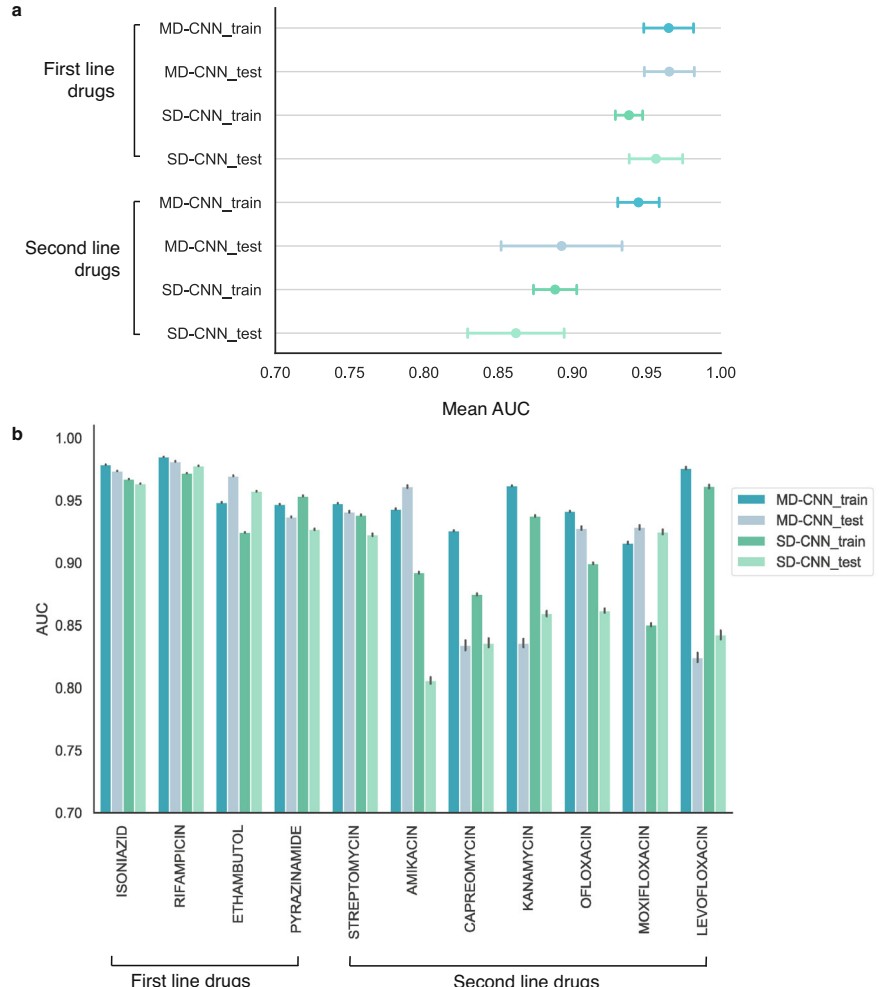

**Fig. 3 MD-CNN and SD-CNN model generalize well on held-out test data.** Performance of CNN models trained on the entire training dataset evaluated on either the entire training dataset or the entire hold-out test dataset ($N = 12{,}848$ isolates). **a** data are presented as mean AUCs ± 95% confidence intervals for first-line (4 drugs) and second-line drugs (7 drugs). **b** AUC for each drug evaluated on either the entire training or entire hold-out test dataset. Ciprofloxacin and ethionamide (both second-line drugs) were not assessed due to low numbers of resistant isolates.

for each nucleotide site in each input sequence using DeepLIFT[24]. For any input, DeepLIFT calculates the change in predicted resistance relative to a reference input, and then backpropagates that difference through all neurons in the network to attribute the change in output to changes in the input variable. We use the pan-susceptible H37Rv genome as the reference input[25]. We take the highest magnitude (positive or negative) saliency score for each nucleotide across all isolates in the training set (see the "Methods" section).

We find evidence that the MD-CNN achieves high performance by relying on drug–drug resistance correlations. Due to the global standard therapeutic regimen for tuberculosis, resistances to first-line drugs almost always evolve before resistances to second-line drugs, and frequently evolve in a particular order[26] (Fig. 4a, b). When considering the top 0.01% ($N = 17$) of positions with the highest DeepLIFT saliency scores for each drug, we observe that an average of 85.0% are known to confer resistance to any drug[27], but only a mean of 24.0% are known to confer resistance to the particular drug being investigated. For example, the top three hits for the antibiotic kanamycin are, in order, a causal hit to the *rrs* gene, an ethambutol-resistance-causing hit to the *embB* gene, and a fluoroquinolone-resistance-causing hit to the *gyrA* gene (Supplementary Data 2).

To probe this further, we introduce mutations that confer resistance to the first-line drugs rifampicin and isoniazid into a pan-susceptible genomic sequence background, in silico, and assess the model predictions for these mutated isolates. The mutations increase the MD-CNN predicted resistance probabilities of pyrazinamide, streptomycin, amikacin, moxifloxacin and ofloxacin resistance (Fig. 5a). The MD-CNN model generalizes well for all five of these drugs: AUC of 0.939 for these drugs, versus 0.831 for the remaining second-line drugs. Taken together, these observations show that the MD-CNN benefits from the correlation structure of antibiotic resistance.

**SD-CNN saliencies highlight known and new potential predictors of resistance**. We assess whether the DeepLIFT saliency scores for the SD-CNN models are able to capture known causal, resistance-conferring variants by cross-referencing the WHO catalog of established resistance-conferring mutations[27] (Supplementary Data 3). We find that, of the 0.1% of sites with the largest absolute DeepLIFT saliencies in each model, a large proportion are in the WHO catalog of known, resistance-conferring positions (ranging from 44.4% for isoniazid to 100% for ofloxacin, see the "Methods" section, Supplementary Table 5). In total, we identify 18 SNP sites in the top 0.1% of sites that are not

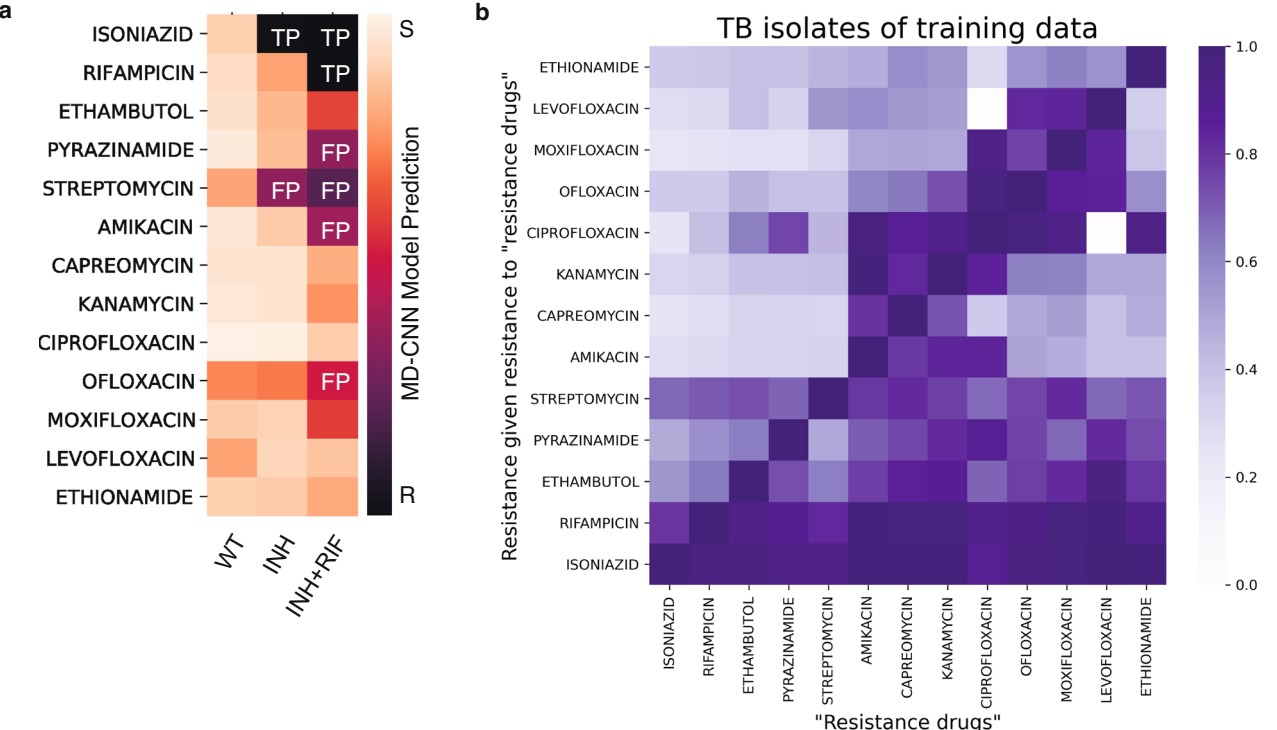

**Fig. 4 MD-CNN learns dependency structure of antibiotic resistance. a** Introduction of single resistance-conferring mutations into pan-susceptible wild-type background (H37Rv, "WT") is sufficient to cause the MD-CNN model to predict false positive resistances. A single isoniazid-resistance conferring mutation (2155168G, "INH") or one isoniazid- and one rifampicin-resistance conferring mutations (2155168G and 761155T, "INH+RIF") are introduced in silico into the wild-type background sequence and resistances are predicted using the MD-CNN model. **b** Dependency heatmaps of drug resistance for training isolates. The horizontal axis represents the drugs to which isolates exhibit resistance. Based on this condition of resistance, the proportions of resistance to other drugs (vertical axis) are computed.

previously known to cause resistance, or are classified by the WHO as of "uncertain significance" (Supplementary Table 6).

We then determine whether the high saliency positions may be indicative of *M. tuberculosis* population structure using pre-determined sets of 62 and 95 lineage-defining genetic variants[28,29], respectively, and variants in perfect linkage with a lineage-defining variant (see the "Methods" section). Lineage or lineage-linked variants comprise only a small proportion of the most salient positions, ranging from 0% to 8% of the top 0.1% of hits for each locus (Supplementary Table 7).

We examine the distribution of saliency scores closely for two drugs with well understood resistance mechanisms: rifampicin and isoniazid; and for pyrazinamide, a drug for which elucidating resistance mechanisms has been more challenging.

*Rifampicin*. Positions in the *rpoB* gene known to cause rifampicin resistance[27] constitute 86% of the top 0.1% and 55% of the top 1% of importance scores (Supplementary Fig. 4). Four of the five highest-scoring variants that have not been previously identified as resistance-causing are located in three-dimensional proximity (minimum atom distance < 8 Å) to resistance-conferring variants in the RpoB protein structure, demonstrating the biological plausibility for these newly identified sites to confer resistance (Supplementary Fig. 4). We identify a three-base-pair insertion at position 761095, RpoB codon L430, as among the top 1% of importance scores. Substitutions at this position, L430P and L430R, are recognized as resistance conferring[27] but insertions are not well characterized.

*Isoniazid*. The common causal site KatG S315 has the highest maximum saliency in the isoniazid SD-CNN (Fig. 5a). We observe several high saliency peaks in the promoter region of the

*ahpC* gene, which are currently designated as of uncertain significance to isoniazid resistance by the WHO[30]. We observe three saliency peaks in the InhA protein, the mycolic acid biosynthesis enzyme targeted by isoniazid. One peak was at the known resistance-conferring mutation S94, and two at positions I21 and I194, of uncertain significance in the WHO catalog. All three of these positions are close in 3D structure (minimum atom distance <8 Å) to the bound isoniazid molecule[31] (Fig. 5b).

*Pyrazinamide*. Of the top 1% of high saliency positions, 85% are known to be resistance-conferring, and an additional 9% are in *pncA*, but not previously known to cause resistance. The top three of these unknown *pncA* mutations are physically adjacent to known resistance-conferring mutations (Fig. 5c, d). The top 1% of salient positions also includes positions in *clpC1*, a gene recently implicated in pyrazinamide resistance, but mutations thereof are not yet recognized to be useful for resistance prediction[32,33] (Supplementary Data 3).

**New potential predictors segregate between resistant and sensitive isolates.** To further validate the 18 high-saliency positions not previously designated as resistance-conferring, we ask whether they are unevenly distributed between resistant and sensitive isolates using the hold-out CRyPTIC dataset. Indeed, 13 of 18 positions have minor alleles that segregate between resistant and sensitive isolates (defined as >80% resistant isolates among isolates with the minor allele). In particular, of all occurrences of minor alleles in the previously highlighted InhA I21 and I194 positions, 100% and 91%, respectively, are found in resistant strains. Of the top three unknown mutations in *pncA*, G78, T100, and R140, 94%, 90%, and 88%, respectively, are found in resistant strains.

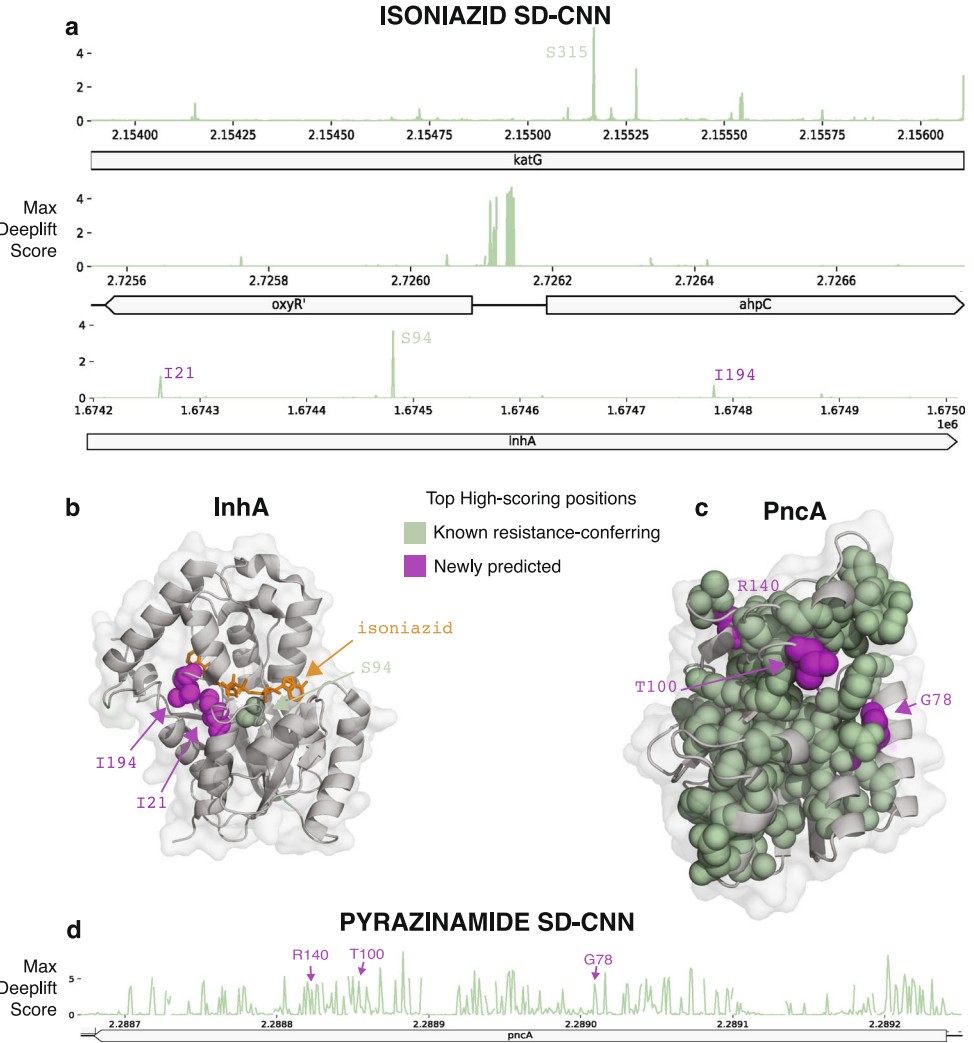

**Fig. 5 SD-CNN saliency scores highlight known and plausible new resistance-conferring loci.** Variants not known to cause resistance according to the WHO[27] are shown in purple. **a** Maximum of absolute value DeepLIFT saliency scores for the isoniazid SD-CNN across all isoniazid-resistant loci. **b** High-importance variants in the InhA protein mapped to its crystal structure[72]. **c** High-importance variants in the PncA protein mapped to its crystal structure[73]. **d** Maxima of absolute value DeepLIFT saliency scores for the pyrazinamide SD-CNN in the *pncA* locus.

## Discussion

In summary, we find that the CNNs offer similar AUCs to the state-of-art WDNN while also being able to discover new loci implicated in resistance, and to visualize them in their genomic context. Although the CNNs have comparable AUCs to the WDNN, the MD-CNN achieves highest sensitivity and the SD-CNN achieves highest specificity for resistance diagnosis. A major advantage of the CNNs is that they predict directly from alignments of genomic loci, allowing the models to consider not only single-nucleotide polymorphisms but also sequence features such as insertions and deletions. They also circumvent challenges arising from differing variant-naming conventions, and reconciling variant features across datasets and time. In addition, applying sequence filters in the convolutional layers allows for pooling of variant effects, such that any deviation from wild-type sequence structure can be more directly associated with resistance even when individual variants are rare. Deep learning methods can also theoretically learn arbitrary interaction terms between input genetic variants and this may further explain performance gains over simpler models like logistic regression. Further study, perhaps using in silico mutagenesis, is required to determine which, if any, epistatic interactions are captured by these models. In this study the detection of epistasis is limited by the use of

binary resistance phenotype data, which masks any epistatic effect that further increases resistance of an already resistant strain. It is hence possible that deep learning models may show further performance improvements over regression methods when applied to quantitative resistance data, where epistatic relationships between variants may be more apparent.

Examining performance by drug, we find the MD-CNN's AUCs to be similar to those of the drug-specific SD-CNNs for first-line drugs, and significantly higher for second-line drugs. CNNs generalize well to the hold-out test isolates for these first-line antibiotics, a promising aspect if they are to be deployed in clinical practice. By contrast, there are more mixed results and generally lower hold-out test AUCs for second-line drugs. For both first-line and second-line drugs, we observe that false negative isolates are often genetically identical at the considered loci to their drug-sensitive counterparts in the training dataset, indicating that additional genetic information is needed to accurately predict the phenotype for certain isolates.

We compare the sensitivity and specificity of our deep learning methods to the current WHO catalog-based method, and find that the catalog method has lower sensitivity. This is not surprising as the catalog was designed to be very selective about which variants are labeled as resistance-conferring, and hence is

much more conservative in calling resistance, highlighting the need for machine learning methods. Machine learning methods also have tunable thresholds that can be optimized to favor sensitivity or specificity depending on the application, a feature that is absent in catalog-based methods.

Our analysis of sensitivity and specificity shows that the MD-CNN has the highest sensitivity of any model analyzed, but that the sensitivity comes at the expense of lowered specificity. This suggests that the MD-CNN, while highly sensitive, has a higher rate of false positive resistance calls, possibly due to the use of the drug–drug correlation structure by the model. In contrast, the SD-CNN has no access to the drug–drug correlation structure and demonstrates the greatest specificity. It may be useful to use both types of CNNs sequentially in clinical settings: if the MD-CNN predicts susceptibility to a drug, the SD-CNN is not consulted; if the MD-CNN predicts resistance to a drug, resistance can be confirmed or disputed through a second prediction from the SD-CNN.

To investigate whether the correlation structure of drug resistance would limit generalizability of the MD-CNN, we generate a dataset with realistic resistance proportions by down-sampling the hold-out CRyPTIC dataset. We find that both deep learning models perform as well or better with samples of realistic resistance proportions, compared to the entire CRyPTIC dataset, particularly in terms of specificity. We propose that the correlation structure of drug resistance is a useful feature for achieving high sensitivity models, and is not problematic in datasets with lower resistance rates. However, should antibiotic usage guidelines change or isolates with unusual resistance patterns be sequenced, the models will require revisiting to ensure that correlation structure does not become a hindrance.

Though neural networks are often criticized for lack of interpretability, we undertake in-silico mutagenesis experiments to understand the behavior of our network. By computationally introducing resistance-conferring mutations into known susceptible sequences, we discover that the MD-CNN's predictions for second-line drugs rely on the correlation structure of drug resistance, which is present in both the training and test sets. Such correlations have been previously observed and shown to improve molecular diagnostic accuracy in *M. tuberculosis*[34]. Indeed, these correlations appear to improve the MD-CNN's sensitivity when predicting resistance to second-line drugs.

We further interpret the behavior of our neural network by assessing DeepLIFT importance scores for every input site. In addition to highlighting known resistance-conferring mutations, our model discovers 18 resistance variants previously unknown or of "uncertain significance" based on the WHO catalog[27]. Including these mutations in resistance prediction may be useful for clinical diagnosis of antibiotic resistance—for example, 6% of isoniazid-resistant strains contain at least one mutation of uncertain significance, and 2.4% contain only mutations of uncertain significance and no canonical resistance variants. The interpretable, nucleotide-level saliency scores permit the protein contextualization of mutations and offer the prospect of modeling how certain mutations would impact protein structure, and drug binding. This can allow for prioritization of putative mutations for further experimental validation.

Limitations of this study include: first, the genomic variants highlighted by saliency analysis and protein contextualization require in silico and in vitro corroboration, although further validation in independent CRyPTIC data supports a causal role. Second, traditional laboratory-based susceptibility testing can have high variance, especially for second-line drugs, introducing a potential source of error. Third, there is insufficient phenotypic data for certain anti-TB drugs (e.g., second-line agents like ethionamide). Finally, additional computational resources would allow the inclusion of more loci of interest, likely augmenting the performance of the MD-CNN and SD-CNNs.

This study demonstrates the feasibility of interpretable, convolutional neural networks for prediction of antibiotic resistance in *M. tuberculosis*. Greater interpretability, reliability, and accuracy make this model more clinically applicable than existing benchmarks and other deep learning approaches. Mapping saliency scores and protein contextualization also offer the possibility of creating hypotheses on mechanisms of anti-TB drug resistance to focus further research. Along with increasingly accessible WGS-capable infrastructure globally, machine-learning-based diagnostics may support faster initialization of appropriate treatment for multi-drug-resistant TB, reducing morbidity and mortality, and improving health economic endpoints[1,35].

## Methods

**Sequence data**. The training, cross-validation, and test datasets consist of a combined 23,049 *M. tuberculosis* isolates for which whole genome sequence data and antibiotic resistance phenotype data are available. The sequencing data are obtained through the National Center for Biotechnology Information database, PATRIC, and published literature,: 10,201 strains are in the "train" dataset (for training and cross-validation)[6,36–48], 7537 are in the hold-out "test_1" dataset (for hold-out testing)[37,49–53], and the remaining 5312 are in the hold-out "test_GenTB" dataset (for hold-out testing)[37,49–53]. Isolates were added to the "test_1" and "test_GenTB" datasets on a rolling basis—i.e., as the sequencing data became available gradually over time.

We process sequences in the train and test_1 datasets using a previously validated pipeline as described by Ezewudo et al. (2018), with modifications as elaborated by Freschi et al. (2021)[48,54]. Reads are trimmed and filtered using PRINSEQ[55], contaminated isolates are removed using Kraken[56], and aligned to the reference genome H37Rv using BWA-MEM[25,57]. Duplicate reads are removed using Picard[58], and we drop isolates with <95% coverage of the reference genome at 10× coverage.

For the "test_GenTB" dataset, we use the sequencing data prepared by Groschel et al.[21], which employs a different variant of the Ezewudo et al. pipeline. The differences between these two pipelines (most notably the use of minimap-2 instead of BWA-MEM) make a negligible difference on final variant calls[59].

With regard to curated genetic variants, the predictor sets of features for the multi-drug wide and deep neural network (WDNN, see the section "Machine learning models" below) are processed as described by Chen et al. (2019)[11]. Conversely, for the single-drug and multi-drug convolutional neural networks (SD-CNN and MD-CNN, see the section "Machine learning models" below), only the FASTA files for the loci of interest are necessary.

**Antimicrobial resistance phenotype data**. Culture-based antimicrobial drug susceptibilities to at least one of 13 anti-TB drugs are available for all 23,049 isolates in the combined training, cross-validation, and test dataset. Phenotypes (drug susceptibility test results) for isolates in the training and cross-validation dataset are from the ReSeqTB data portal, the PATRIC database, and manual curation of phenotypic data available in the literature[3,6,36–48,60]. Phenotypes for the test dataset isolates are from data available in the literature[37,49–53]. Each isolate's phenotype is classified as resistant, susceptible, or unavailable, with respect to a combination of 13 possible first-line (rifampicin, isoniazid, pyrazinamide, ethambutol) and second-line drugs (streptomycin, ciprofloxacin, levofloxacin, moxifloxacin, ofloxacin, capreomycin, amikacin, kanamycin, ethionamide). (Table 1). In the hold-out test dataset, ethionamide and ciprofloxacin are excluded due to having fewer than 50 phenotyped resistant isolates (0/2 resistant to ciprofloxacin; 12/25 resistant to ethionamide) (Table 2).

**Selecting input loci**. The loci of the isolate sequences are selected from genes known or suspected to cause resistance based on previous models and experiments (Table 3). In order to incorporate any possible regulatory sequences from the immediate genetic neighborhood, the entire upstream and downstream region of each gene or operon is included (upstream region: from the beginning of the relevant gene to the end of the previous gene on the genome; downstream region: from the end of the relevant gene to the beginning of the next gene on the genome). Loci are aligned to the H37Rv reference genome for comparison of coordinates and genome annotations are based on H37Rv coordinates from Mycobrowser[61].

**Machine learning models**. The multi-drug (multi-task) wide-and-deep neural network (WDNN) is described by Chen et al. (2019), and involves three hidden layers (256 ReLU), dropout, and batch normalization[11]

The multi-drug convolutional neural network (MD-CNN) comprises two convolution layers (with filter size 12 nucleotides in length), one max-pooling layer, two convolution layers, one max-pooling layer, followed by two fully connected

hidden layers each with 256 rectified linear units (ReLU) (Fig. 1). This architecture is selected based on its performance, as defined by area under the receiver operator characteristic curve (AUC), compared to other architectures with fewer convolutional layers and different filter sizes (Supplementary Fig. 1). Neither random nor cartesian grid search of optimal hyperparameters is conducted.

The MD-CNN is trained for 250 epochs via stochastic gradient descent and the Adam optimizer (learning rate of $e^{-9}$). We select an optimal number of epochs based on minimizing validation loss (Supplementary Fig. 5). The training is performed simultaneously using the resistance phenotype for all 13 drugs, hence the 13 nodes in the final output layer, the output of each node corresponding to the sigmoid probability of the strain being resistant to the respective drug.

The MD-CNN's loss function is adapted from the masked, class-weighted binary cross-entropy function described by Chen et al. (2019)[11]. This function addresses the dataset imbalance (missing resistance phenotypes for a varying number of drugs in any given isolate) by upweighting the sparser of the susceptible and resistant classes for each drug, and masking outputs where resistance status was completely missing.

The single-drug convolutional neural networks (SD-CNNs) are 13 individually trained convolutional neural networks, each trained to predict for only one drug, hence the output layer having size one instead of 13. Each SD-CNN is given only the input loci relevant to its particular antibiotic, resulting in different input sizes depending on the longest locus for each drug. The architecture for the SD-CNNs is otherwise identical to that of the MD-CNN. The SD-CNNs are initially trained for 150 epochs using stochastic gradient descent and the Adam optimizer (learning rate of $e^{-9}$), and an optimal number of epochs for each SD-CNN is selected to minimize the validation loss.

**Logistic regression benchmark**. We build a logistic regression benchmark to evaluate the performance of our neural network models. For each of the 18 input loci used in the MD-CNN and SD-CNNs, we select all sites with a minor allele frequency of at least 0.1%, resulting in 3011 sites across 23,049 genomes. Sites are then encoded using a major/minor allele encoding.

Using the same train/test partitioning as for the neural network models, we use GridSearchCV in Scikit-learn v.0.23.2[62] to select the optimal L2 penalty weight for a Logistic Regression classifier with balanced class weights. Hyperparameter search is performed for each drug independently, testing the values $C = [0.0001, 0.001, 0.01, 0.1, 1]$. After selecting the optimal L2 weight, we use five-fold cross-validation on the training set to assess the AUC, specificity, and sensitivity, selecting a model threshold that maximizes the sum of specificity and sensitivity.

**Training and model evaluation**. Five-fold cross-validation is performed five times to obtain the performance metrics—area under the receiver operator characteristic curve (AUC), sensitivity, specificity, and probability threshold (to maximize the sum of sensitivity and specificity)—and the 95% confidence intervals of the AUC values between the models.

Model performance on the hold-out test sets is evaluated using the probability threshold selected during training.

**Computational details**. The MD-CNN is developed and implemented using TensorFlow 2.3.0 in Python 3.7.9 with CUDA 10.1[63–65]. Model training is performed on an NVIDIA GeForce GTX Titan X graphics processing unit (GPU).

**Evaluation on CRyPTIC isolates**. Binary phenotype data is downloaded from the CRyPTIC study[22]. CRyPTIC has phenotype data for the following drugs predicted by the CNN: isoniazid, rifampicin, ethambutol, amikacin, kanamycin, moxifloxacin, levofloxacin, ethionamide. Phenotypes whose quality is not "high" are masked. We filter isolates that do not have a phenotype for any of the drugs of interest, for a total of 9498 isolates.

Isolates from the dataset of 9498 are analyzed using a variant of the Ezewudo et al. pipeline, with modifications as elaborated by Freschi et al. (2021)[48,54], additionally using minimap2-2.24 for read mapping, SPAdes v 3.15.4 for assembly, and trimmomatic v. 0.40 for read trimming[66–68]. Nucleotide sequences for the designated genomic loci are extracted and aligned against the previous input sequence alignments using MAFFT v7.490 with the --add and --keeplength options[69].

Confidence intervals for sensitivity and specificity on the full CRyPTIC dataset are generated by sampling 80% of the dataset 100 times.

**Analysis of mis-predicted isolates**. For each SD-CNN model, we compute the genetic distance (number of different sites) between all isolates in the training and test sets. Following Vargas et al., 2021[70], to compute the genetic distance for the entire genome, we process the VCF files output by our pipeline described above, and take all sites that meet the following criteria: mean Base Quality > 20, mean Mapping Quality > 30, no reads supporting insertions or deletions, number of high quality reads ≥ 20, and at least 75% support for a non-reference allele. Sites falling between 25% and 75% support for non-reference alleles are labeled as uncertain and do not contribute to distance calculation. We further remove sites with an empirical base pair recall score < 90%[59] and sites where at least 10% of the isolates

have uncertain calls. We then compute the number of differences between pairs of isolates.

For computing the number of differences from the perspective of the SD-CNN model, only sites found in the loci used in each SD-CNN model are included, and each site with a confident insertion or deletion contributes one to the difference score.

**Designation of known resistance variants from WHO catalog**. A list of known resistance-conferring variants is extracted from the WHO catalog[27]. Only variants with a Final Confidence Grading of "Category 1: Associated with resistance" or "Category 2: Associated with resistance—interim" are taken to be known resistance-conferring variants.

For prediction of resistance with the WHO catalog, isolates are assumed to be sensitive unless they have one or more of the known resistance-conferring variants for a particular drug.

**Saliency calculation**. Saliencies are calculated using DeepLIFT v. 0.6.12.0, using the recommended defaults for genomics: "rescale" rule applied to convolutional layers, and "reveal-cancel" rule applied to fully connected layers. We use the H37Rv reference genome, which is sensitive to all antibiotics, as the baseline[25].

Saliency scores for each isolate sequence are calculated relative to the H37Rv baseline. For our analysis of positions influencing antibiotic resistance prediction, we take the maximum of the absolute value of the scores at each position across all resistant isolates.

**Lineage variant analysis**. We define lineage variants as those found in the Coll et al. or Freschi et al. barcode of lineage-defining variants[28,29]. We further annotate any position in our 18 loci as lineage-associated if that position has an identical distribution of major/minor alleles to any position in the Freschi et al. barcode, excluding the position *1,137,518* which defines lineage 7 (not present in our dataset).

**Reporting summary**. Further information on research design is available in the Nature Research Reporting Summary linked to this article.

## Data availability

All code, processed input data, and saved model files are available on github, https://github.com/aggreen/MTB-CNN: v1.0[71]. The processed strain phenotype data used in this study are available in MTB-CNN/input_data/master_table_resistance.csv and MTB-CNN/input_data/cryptic_phenotype_data.csv. The raw read data are publicly available for download from the NCBI using accession codes found in the processed strain phenotype data files. The processed FASTA files used as input to the CNNs are available in MTB-CNN/input_data/fasta_files and MTB-CNN/input_data/cryptic. The in silico mutagenized strains are available in MTB-CNN/input_data/dummy_strain_fasta_files. The trained MD-CNN and SD-CNN models are available in MTB-CNN/saved_models. The model evaluation statistics generated in this study are provided in Supplementary Tables 1–4. The saliency score data generated in this study are provided in Supplementary Data 2 (MD-CNN) and Supplementary Data 3 (SD-CNNs). Summaries and analysis of the saliency score data generated in this study are available in Supplementary Tables 5–7.

## Code availability

Implementation of all models and data analysis can be found at: https://github.com/aggreen/MTB-CNN: v1.0[71].

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

## Acknowledgements

We thank members of the Farhat lab for discussion and input. We are grateful to Dr. Peter Koo, Dr. Avika Dixit, Dr. Payman Yadollahpour, Greg Raskind, and Jiqing Zhu for discussions regarding saliency score calculation, validation analyses on the WDNN, CNN codebase proofreading, and CRyPTIC isolate phenotypes. Computational resources and support were provided by the Orchestra High Performance Compute Cluster at Harvard Medical School, which is funded by the NIH (NCRR 1S10RR028832-01). A.G.G. was supported by a National Institutes of Health NLM Training Grant T15LM007092 and NIH/NIAID F32AI161793. C.H.Y. was supported by the US–UK Fulbright Commission (USA/UK), the BUNAC Educational Scholarship Trust (UK), the Gavin and Ann Kellaway Research Fellowship (Auckland Medical Research Foundation, New Zealand), and the Royal Australasian College of Physicians Rowden White Fellowship (Australasia). M.I.G. was supported by the German Research Foundation (GR5643/1-1). M.F. is supported by NIH/NIAID R01AI155765.

## Author contributions

M.R.F., A.B., A.G.G., C.H.Y., and M.L.C. conceived the study and designed the analyses. A.G.G, C.H.Y., and M.L.C. implemented the CNN code. A.G.G. and C.H.Y. performed the analyses. M.R.F. and A.B. supervised the research. A.G.G., C.H.Y., M.R.F., and A.B. wrote the manuscript. Y.E., M.F., L.F., and M.I.G. contributed data and discussed data processing. All authors reviewed the manuscript.

## Competing interests

The authors declare no competing interests.
