## [Peer Review File · Nature Communications]

Reviewers' Comments:

Reviewer #1:

Remarks to the Author:

In this study, Green and colleagues develop a deep convolutional neural network (CNN) able to predict drug resistance from Mtb DNA sequences. Major strengths of this work include: (1) the large dataset used (>20,000 samples); (2) source data and code are made available, facilitating reproducibility and adoption of methods; (3) methods to interpret CNNs are applied, resulting in identification of salient features. See major and minor comments below.

Major comments:

1. About the benchmark model used. The authors use a multi-drug wide-and-deep neural network (MD-WDNN) and logistic regression with L2 regularization penalty as model benchmarks. The authors need to be more explicit about the rationale for using these models as benchmark, and the use of MD-WDNN as the "state-of-the-art", as this is an important claim. They cite their previous work (citation 11, Chen et al. 2019), where they tried a variety of statistical/machine learning techniques to conclude that "The highest performing simple and complex models were L2 regularized logistic regression and the MD-WDNN trained on the full predictor set, respectively". I believe the authors should also attempt to "benchmark" their models against "catalogue-based" approaches including Mykrobe, TB-profiler or the latest WHO catalogue of established resistance-conferring mutations, as these are (or may be expected to be) widely employed by the community.

2. The performance of models is expressed as mean AUC. It is also common practice in the field to report the diagnostic performance in terms of sensitivity and specificity compared to phenotypic DST. The authors should also present a summary of predictive performance for all models tested (as presented in Supplementary Table 1) in terms of sensitivity and specificity. Also, In Supplementary Table 1, please provide a full list of abbreviations and include the mean AUC for each model to facilitate interpretation.

3. I am finding hard to tease apart the improvement in performance attributable to the underlying machine learning algorithm (i.e. WDNN vs. CNN) from that attributable to the underlying genetic features (SD vs. MD) from the results presented in section "Benchmarking CNN models against state-of-the-art". From statements like "Against the state-of-the-art WDNN, the AUCs, sensitivities, and specificities of the MD-CNN are comparable: the MD-CNN's mean AUC is 0.948 (vs. 0.959 for the MD-WDNN, $q=0.15$) for first-line drugs, and 0.912 (vs. 0.924 for the MD-WDNN, $q=0.30$) for second-line drugs", it seems the machine learning algorithm does not make a difference. From the statement: "The SD-CNN (mean AUC of 0.938 for first-line drugs; mean AUC of 0.877 for second-line drugs) performs comparably to the MD-CNN for first-line drugs ($q=0.19$), and is less accurate than the MD-CNN for second-line drugs ($q=0.009$).", it seems the improvement in performance of MD-CNN is attributable to the underlying features.

4. The analysis performed to investigate the source of "missed resistance (false negatives)" is relevant. The authors interpret that finding pairs of isolates being genomically identical at all 18 candidate loci, one being a false negative and the second being sensitive, points to additional novel resistance mutations outside the 18 candidate loci considered. Did the authors check the genome-wide genetic distance too? In other words, are the false negative isolates having a genomically identical yet sensitive isolate identical or similar at the whole-genome level too? If so, that could point also to phenotypic testing issues (technical errors or uncertainty around the MIC breakpoints) as a source of false negatives too.

5. One of the main conclusions is that MD-CNN achieves higher predictive performance by using drug resistance correlations. This improvement is particularly important for second-line drugs. As the authors note, "Due to the global standard therapeutic regimen for tuberculosis, resistances to first-line drugs almost always evolve before resistances to second-line drugs". Do the authors think that using all 18 candidate loci to predict resistance to individual drugs entails an increased risk of calling false positives? The authors indeed show in Figure 4A that well-known isoniazid and rifampicin resistance mutations cause the MD-CNN model to predict false positive resistances.

Here, as pointed in a previous comment, presenting the predictive performance of SD-CNN vs. MD-CNN in terms of specificity and sensitivity would be informative to clarify this point. Could the authors also elaborate on the limitations that CNN's predictions for second-line drugs rely on the correlation structure of drugs?

Minor comments:

If journal's formatting guidelines allow, please extend the abstract to include more details about main the findings of this work. At the moment, statements in the abstract are rather generic.

The dataset analysed in this study (23,049 M. tuberculosis isolates) is obtained from a variety of published sources, including the NCBI, PATRIC and published literature. Given the diversity of sources, and to facilitate reproducibility, the authors should make sure the genome accession, source study (e.g. PubMed ID), and antibiogram of each isolates are made available as a supplementary data file as part of this manuscript. Source data seems to have been uploaded on GitHub (file master_table_resistance.csv) but this needs to be referenced in the main manuscript, and all genome accessions included.

I couldn't find a justification as to why the "test_GenTB" dataset was analyzed using a different variant calling pipeline.

The authors state that "All drugs are represented by at least 250 phenotyped isolates." But did the authors consider a minimum proportion of resistance isolates as a drug exclusion criterium? The authors mention that some drugs "contained low resistance counts" and were not considered, but the threshold used is not reported in the manuscript.

The authors mention that the hold-out dataset was curated on a rolling basis. The authors should explain what "rolling curation" means in the methods section.

At some point the authors refer to "Variants associated with the M. tuberculosis population structure". However, I couldn't find how these were defined or extracted from.

The authors indicate that an advantage of CNN is that "features such as insertions and deletions or more complex variation" are considered by the model. Did the authors find any indels among salient features?

The authors indicate that "6% of isoniazid resistant strains contain at least one newly discovered mutation, and 2.4% contain only newly discovered mutations and no canonical resistance variants.". What do the authors mean by "newly discovered mutations"? those present in the WHO "uncertain significance" category? or those never reported in the literature? The authors should be careful not to refer to mutations absent in this category as "newly discovered mutations" as rather stringent statistical criteria were applied to classify mutations into "Group 1: Associated with resistance" and "Group 2: Associated with resistance – interim" categories.

In Table 1, or in a new Supplementary Table, please indicate the coordinates of the putative regulatory regions upstream from genes.

In statement "The Multi-Drug Convolutional Neural Network (MD-CNN) comprises two convolution layers (with filter size 12 nucleotides in length), one max-pooling layer, two convolution layers, one max-pooling layer, followed by two fully-connected hidden layers each with 256 rectified linear units (ReLU) (Table 1)". Do the authors want to refer to Figure 1 or Table 1?

Given the increasing appreciation that drug resistance in TB is a multi-locus phenotype (as shown by recent work by this group on epistatic interactions) do the authors think their CNN models can capture these epistatic relationships systematically? Or would it be confounded by common drug resistance co-occurrences?

Reviewer #2:

Remarks to the Author:

In their paper Green, Yoon and co-workers present a deep convolutional neural network that predicts the antibiotic resistance phenotypes of *Mycobacterium tuberculosis* isolates. The authors have trained and cross-validated data from more than 10000 *M.tuberculosis* strains from ReSeqTB and the WHO Supranational Reference laboratory network, whereby each isolate was phenotyped for resistance to at least one of 13 anti-TB drugs, and wherein all drugs are represented by at least 250 phenotypes isolates.

The authors find that their presented convolutional neural networks offer similar predictive accuracy to the state-of-art prediction method (published by them some time ago), while being able to identify new potential resistance mutations.

This is particularly visible in Fig. 5 where the authors depict some of newly predicted mutations in gene products of *inhA* and *pncA*, which are of uncertain significance in the WHO catalogue, that are likely involved in resistance to Isoniazid and Pyrazinamide, respectively.

These are very important outcomes of this analyses and the authors argue that these mutations are probably involved in resistance, as their positions are very close to the active sites in 3 D models, similar to known resistance mutations.

However, it would still be very helpful for the validation of the predictions if the authors showed some confirmation of their predictions, for example by highlighting strains in global *Mtb* databases that show such mutations and are reportedly resistant to the respective anti-TB drugs.

Alternatively, the authors might also take in consideration potential collaborations with wet-lab mycobacteriologists that could produce mycobacterial strains with the proposed mutations and see whether these had indeed some impact on respective anti-TB resistance characteristics, and/or try to raise resistant mutants against the respective drugs and see if the predicted mutations might be part of the mutant spectrum observed.

Such data would make the manuscript less speculative than it is at the current stage where the predictions have not been confirmed by literature or experimental data. It is true that the later confirmation steps would need a substantial amount of time, especially for slow growing mycobacteria, but may be the authors could also find alternative, faster models to show these links.

Reviewer #3:

Remarks to the Author:

This work proposes a DNN that provides interpretability. By doing so they achieve similar performance to previous NN publications but provide insights on what is learned.

The work is performed and presented state-of-the-art. Below are some comments that may be of interest to the authors.

1. While it is true that classic drug resistance testing takes a long time, there is a good implementation of point of care detection of drug resistance that serves the purpose of detecting resistance to rifampin that triggers a different pattern of care. It is not here that the problem resides, and where genome sequencing and DNN are going to change practice. Rather, it is the fine dissection of complex drug resistance that may benefit, in a second step, the long term treatment decisions.
2. The observation that there is a drug-drug correlation structure may need more text. It works in this setting because the training (and testing datasets) are enriched for resistance and multidrug resistance may be a significant obstacle to generalizing the model to the real world pattern of resistance in the general population infected with conventional circulating strains. One possible approach would be to test the models against a testing set that reflects the patterns in the general testing laboratories. Otherwise, a model trained on enriched datasets may only be fit for a reference laboratory that tests high numbers of resistant, MDR and extended MDR strains.
3. The manuscript correctly implements concepts of circulating lineages and how much this genetic information contributes to the results. However, this is an important potential confounder and may need more text to reassure the reader that this is adequately controlled for.

4. The authors may use some collapsing of drug class members to enhance the learning. For example, all fluoroquinolones can be collapsed together because they are felt to share to large extent the mechanisms of resistance.
5. The identification and mapping to protein structures of candidate resistance variants is a very important contribution of this study. Thus, the value of the work would be maximized if the authors tested a couple of the proposed variants in vitro. It is clear that this is not done overnight – but it would be great if this could be included in the manuscript.
6. Last point refers to clinical communication of the results. AUCs are the workhorse of this type of work. However, clinical laboratories and medical staff use the concepts of sensitivity, specificity, pos predictive value etc. It would be useful to speak to this audience presenting a reasonable translation of the performance of such tools in the clinical laboratory setting.

Reviewer #1 (Remarks to the Author):

In this study, Green and colleagues develop a deep convolutional neural network (CNN) able to predict drug resistance from Mtb DNA sequences. Major strengths of this work include: (1) the large dataset used (>20,000 samples); (2) source data and code are made available, facilitating reproducibility and adoption of methods; (3) methods to interpret CNNs are applied, resulting in identification of salient features. See major and minor comments below.

Major comments:

Comment 1.1. *About the benchmark model used. The authors use a multi-drug wide-and-deep neural network (MD-WDNN) and logistic regression with L2 regularization penalty as model benchmarks. The authors need to be more explicit about the rationale for using these models as benchmark, and the use of MD-WDNN as the “state-of-the-art”, as this is an important claim. They cite their previous work (citation 11, Chen et al. 2019), where they tried a variety of statistical/machine learning techniques to conclude that “The highest performing simple and complex models were L2 regularized logistic regression and the MD-WDNN trained on the full predictor set, respectively”. I believe the authors should also attempt to “benchmark” their models against “catalogue-based” approaches including Mykrobe, TB-profiler or the latest WHO catalogue of established resistance-conferring mutations, as these are (or may be expected to be) widely employed by the community.*

Response 1.1: We thank the reviewer for their detailed and insightful comments on our manuscript.

We note that the WDNN model has previously been benchmarked against Mykrobe and TB-profiler in a recent paper from our group, finding overall higher sensitivity at the expense of slightly lower specificity for the WDNN versus catalog-based methods (Gröschel et al 2021, DOI: [10.1186/s13073-021-00953-4](https://doi.org/10.1186/s13073-021-00953-4)).

As the reviewer points out, the WDNN and regression+L2 models were found to be the best in the original WDNN paper, and hence were chosen as the benchmarks for our analysis. In addition, we feel these benchmarks are appropriate as the WDNN is a current state-of-art machine learning model, and penalized regression is a standard simple model used to benchmark more complex machine learning techniques. We have clarified these points in the manuscript results section as follows:

We benchmark both types of CNNs against an existing state-of-the-art multi-drug wide-and-deep neural network (MD-WDNN) and a logistic regression with L2 regularization penalty, as these methods were found to perform similarly and outperform a random forest classifier (11). The WDNN was also found to have higher sensitivity than existing catalogue-based methods in a recent comparative study (21).

We have taken the reviewer’s suggestion to compare our model performance against the performance of the recently released WHO catalog of resistance-conferring variants. We find that the catalog-based method has reduced sensitivity compared to our deep learning methods.

This is not surprising as the catalog was designed to be very selective about which variants are labeled as resistance-conferring, and hence is much more conservative in calling resistance, highlighting the need for more sophisticated methods. We have now added discussion of the catalog comparison in the Results sections as follows:

We then compare the sensitivity and specificity of the MD-CNN to the field-standard WHO catalog of known resistance-conferring variants (Methods). In general, we find higher sensitivity for the MD-CNN model versus the WHO catalog (mean sensitivity 91.9% for first-line drugs [MD-CNN] vs. 80.4% [WHO catalog]; 91.1% for second-line drugs [MD-CNN] vs. 73.1% [WHO catalog]) at the expense of lower specificity (92.3% for first-line drugs [MD-CNN] vs. 94.8 % [WHO catalog]; 85.9% for second-line drugs [MD-CNN] vs. 93.6% [WHO catalog]) (Supplementary Table 3).

Comment 1.2 *The performance of models is expressed as mean AUC. It is also common practice in the field to report the diagnostic performance in terms of sensitivity and specificity compared to phenotypic DST. The authors should also present a summary of predictive performance for all models tested (as presented in Supplementary Table 1) in terms of sensitivity and specificity. Also, In Supplementary Table 1, please provide a full list of abbreviations and include the mean AUC for each model to facilitate interpretation.*

Response 1.2: We have taken the reviewer's suggestion to compute and discuss sensitivity and specificity of our models, and have added a new section to the results discussing these comparisons, as well as a new Supplementary Table 2 with side-by-side comparisons of sensitivity and specificity for each model. We have also included the requested list of abbreviations and mean AUCs in Supplementary Table 1.

An important feature of the CNN models is the ability to tune the model threshold to optimize sensitivity or specificity, depending on the application. We choose a threshold for all of our machine learning models (MD-CNN, SD-CNN, logistic regression + L2, and WDNN) that maximizes the sum of sensitivity and specificity. We find that the MD-CNN has the highest sensitivity of the four models for first-line (mean sensitivity 91.9%) and second-line drugs (mean sensitivity 91.1%) except ethambutol, for which the WDNN exhibits the highest sensitivity (Supplementary Table 2). The SD-CNN demonstrates the greatest specificity for first-line drugs (mean specificity 94.1%) except ethambutol where the MD-CNN has the highest; the SD-CNN demonstrates the highest specificity for second-line drugs (mean specificity 94.3%) except ethionamide and ciprofloxacin where the MD-CNN out-performs (Supplementary Table 2).

Comment 1.3. *I am finding hard to tease apart the improvement in performance attributable to the underlying machine learning algorithm (i.e. WDNN vs. CNN) from that attributable to the underlying genetic features (SD vs. MD) from the results presented in section "Benchmarking CNN models against state-of-the-art". From statements like "Against the state-of-the-art WDNN, the AUCs, sensitivities, and specificities of the MD-CNN are comparable: the MD-CNN's mean AUC is 0.948 (vs. 0.959 for the MD-WDNN, $q=0.15$) for first-line drugs, and 0.912 (vs. 0.924 for the MD-WDNN, $q=0.30$) for second-line drugs", it seems the machine learning algorithm does not make a difference. From the statement: "The SD-CNN (mean AUC of 0.938 for first-line drugs; mean AUC of 0.877 for second-line drugs) performs comparably to the MD-*

CNN for first-line drugs ($q=0.19$), and is less accurate than the MD-CNN for second-line drugs ($q=0.009$).”, it seems the improvement in performance of MD-CNN is attributable to the underlying features.

Response 1.3: We believe that the difference in performance between the MD-CNN and SD-CNN is attributable to two factors: the MD-CNN has both a different set of underlying features and a different architecture compared to the SD-CNN. While the MD-CNN has access to 18 genetic loci and predicts resistance data for 13 drugs at once, the SD-CNN predicts only a single drug and has access to only the relevant genetic loci for that particular drug. So, the improvement in MD-CNN performance over SD-CNN may be attributable to both the presence of loci not causally related to but correlated with the drug resistance in question and/or that certain resistance phenotypes share underlying genetic mechanisms. We have expanded our explanation of this in the results section:

*As superior performance of multi-task over single-task models has been demonstrated with convolutional neural networks in computer vision(18–20), the MD-CNN is designed to optimize performance by combining all genetic information and relating it to the full resistance antibiogram. We compare the MD-CNN with 13 single-drug convolutional neural networks (SD-CNN), each of which has a single-task, single-label architecture, in which only loci with previously known causal associations for any given drug are incorporated (**Supplementary Figure 2**). Because the MD-CNN has access to all 18 loci related to any drug resistance, differences in performance may be attributable to both the fact that certain resistance phenotypes share underlying genetic mechanisms, and/or the presence of loci not causally related to but correlated with the drug resistance in question.*

Comment 1.4 *The analysis performed to investigate the source of “missed resistance (false negatives)” is relevant. The authors interpret that finding pairs of isolates being genomically identical at all 18 candidate loci, one being a false negative and the second being sensitive, points to additional novel resistance mutations outside the 18 candidate loci considered. Did the authors check the genome-wide genetic distance too? In other words, are the false negative isolates having a genomically identical yet sensitive isolate identical or similar at the whole-genome level too? If so, that could point also to phenotypic testing issues (technical errors or uncertainty around the MIC breakpoints) as a source of false negatives too.*

Response 1.4. We have undertaken this suggestion and analyzed the full genomes of the false negative isolates from the test dataset, to determine if additional genetic variation exists outside the 18 loci analyzed that could be explaining the missed resistance. We find that for each drug, most (>94%) of the false negative isolates are not identical to any isolate in the training set when the whole genome is considered. This supports our hypothesis that uncharacterized genetic variation outside of the studied 18 loci may explain missed resistance in the false negative isolates. We have elaborated on this experiment in the Results and the corresponding methods section:

Indeed, when considering the entire genome, almost no false negative test isolates are identical to a sensitive isolate in the training set (less than 6% of isolates for all drugs),

indicating that additional genetic variation does exist and may lead to the currently unexplained resistance.

Comment 1.5. *One of the main conclusions is that MD-CNN achieves higher predictive performance by using drug resistance correlations. This improvement is particularly important for second-line drugs. As the authors note, “Due to the global standard therapeutic regimen for tuberculosis, resistances to first-line drugs almost always evolve before resistances to second-line drugs”. Do the authors think that using all 18 candidate loci to predict resistance to individual drugs entails an increased risk of calling false positives? The authors indeed show in Figure 4A that well-known isoniazid and rifampicin resistance mutations cause the MD-CNN model to predict false positive resistances. Here, as pointed in a previous comment, presenting the predictive performance of SD-CNN vs. MD-CNN in terms of specificity and sensitivity would be informative to clarify this point. Could the authors also elaborate on the limitations that CNN’s predictions for second-line drugs rely on the correlation structure of drugs?*

Response 1.5: Our new analysis of sensitivity and specificity shows that the MD-CNN has the highest sensitivity of any model analyzed, but that the sensitivity comes at the expense of lowered specificity (compared to the SD-CNN for example). This supports the reviewer’s suggestion that the MD-CNN, while highly sensitive, is more likely to call false positive resistances, possibly due to the use of the drug-drug correlation structure by the model.

To further examine the generalizability and effects of drug-drug correlation structure, we conducted an additional experiment using our models to predict resistances to the CRyPTIC dataset isolates, where we down-sampled the proportion of resistant isolates to better reflect the prospective application of these models in practice (See **Comment 3.2**). Both models performed slightly better on the down-sampled dataset than on the full CRyPTIC dataset, suggesting we do not expect performance deficits on datasets with lower resistance proportions that are expected in clinical practice in many high TB prevalence settings.

Minor comments:

1.1 *If journal’s formatting guidelines allow, please extend the abstract to include more details about main the findings of this work. At the moment, statements in the abstract are rather generic.*

1.1 We have made the abstract more descriptive of the results of the study within the journal limit of 150 words:

Long diagnostic wait times hinder international efforts to address multi-drug resistance in M. tuberculosis. Pathogen whole genome sequencing, coupled with statistical and machine learning models, offers a promising solution. However, generalizability and clinical adoption have been limited by a lack of interpretability, especially in deep learning methods. Here, we present two deep convolutional neural networks that predicts the antibiotic resistance phenotypes of M. tuberculosis isolates: a multi-drug CNN (MD-CNN), that predicts resistance to 13 antibiotics based on 18 genomic loci, with AUCs 82.6-99.5% and higher sensitivity than state-of-the-art methods; and a set of 13 single-drug CNNs (SD-CNN) with AUCs 80.1-97.1% and higher specificity than the previous

state-of-the-art. Using saliency methods to evaluate the contribution of input sequence features to the SD-CNN predictions, we identify 18 sites in the genome not previously associated with resistance. The CNN models permit functional variant discovery, biologically meaningful interpretation, and clinical applicability.

1.2 *The dataset analysed in this study (23,049 M. tuberculosis isolates) is obtained from a variety of published sources, including the NCBI, PATRIC and published literature. Given the diversity of sources, and to facilitate reproducibility, the authors should make sure the genome accession, source study (e.g. PubMed ID), and antibiogram of each isolates are made available as a supplementary data file as part of this manuscript. Source data seems to have been uploaded on GitHub (file master_table_resistance.csv) but this needs to be referenced in the main manuscript, and all genome accessions included.*

We have updated master_table_resistance.csv for with NCBI genome accessions for all isolates.

1.3 *I couldn't find a justification as to why the "test_GenTB" dataset was analyzed using a different variant calling pipeline.*

The two datasets were analyzed with slightly different pipelines because the data was taken from two different source studies, which had processed the isolates with different pipelines. We have clarified this in the manuscript, and highlighted that the differences between the pipelines has been shown to have a negligible effect on variant calling.

For the "test_GenTB" dataset, we use the sequencing data prepared by Groschel et al.(52), which employed a different variant of the Ezewudo et al. pipeline. The choice of minimap2 as the aligner rather than BWA-MEM has been shown to have a negligible effect on variant calling in M. tuberculosis (53).

1.4 *The authors state that "All drugs are represented by at least 250 phenotyped isolates." But did the authors consider a minimum proportion of resistance isolates as a drug exclusion criterium? The authors mention that some drugs "contained low resistance counts" and were not considered, but the threshold used is not reported in the manuscript.*

1.4 We have clarified in the text that we used a threshold of 50 resistant isolates for test data exclusion in the methods. The drug with the lowest number of phenotyped resistant isolates included was capreomycin, with 61 R isolates.

1.5 *The authors mention that the hold-out dataset was curated on a rolling basis. The authors should explain what "rolling curation" means in the methods section.*

1.5. We have added the following in the methods section to explain:

Isolates were added to the "test_1" and "test_GenTB" datasets on a rolling basis – ie, as the sequencing data became available gradually over time.

1.6 *At some point the authors refer to "Variants associated with the M. tuberculosis population structure". However, I couldn't find how these were defined or extracted from.*

1.6 The description of lineage-conferring variants is found in the methods section:

Lineage variant analysis

We define lineage variants as those found in the Coll et al. or Freschi et al. barcode of lineage-defining variants(60, 61). We further annotate any position in our 18 loci as lineage associated if that position has an identical distribution of major/minor alleles to any position in the Freschi et al. barcode, excluding the position 1,137,518 which defines lineage 7 (not present in our dataset).

We have also expanded our discussion of lineage-conferring variants in the result section (see Comment 3.3):

1.7 *The authors indicate that an advantage of CNN is that “features such as insertions and deletions or more complex variation” are considered by the model. Did the authors find any indels among salient features?*

1.7 We have updated our analysis of the salient features to explicitly include insertions and deletions relative to the reference sequence H37Rv, which are now included in our Extended Data tables of high saliency positions. We highlight an insertion of interest among the high-salient features for rifampicin, as now discussed in the Results section:

We identify a three base-pair insertion at position 761095, RpoB codon L430, as among the top 1% of importance scores. Substitutions at this position, L430P and L430R, are recognized as resistance conferring (24) but insertions are not well- characterized.

1.8 *The authors indicate that “6% of isoniazid resistant strains contain at least one newly discovered mutation, and 2.4% contain only newly discovered mutations and no canonical resistance variants.”. What do the authors mean by “newly discovered mutations”? those present in the WHO “uncertain significance” category? or those never reported in the literature? The authors should be careful not to refer to mutations absent in this category as “newly discovered mutations” as rather stringent statistical criteria were applied to classify mutations into “Group 1: Associated with resistance” and “Group 2: Associated with resistance – interim” categories.*

1.8 We have replaced the phrase newly discovered mutation as “mutation of uncertain significance” in the above text, and clarified in the methods section that our “known resistance variants” are the WHO mutations in Group 1 and Group 2:

Designation of known resistance variants

A list of known resistance-conferring variants are extracted from the WHO catalog (24). Only variants with a Final Confidence Grading of “Category 1: Associated with resistance” or “Category 2: Associated with resistance – interim” are taken to be known resistance-conferring variants.

1.9 *In Table 1, or in a new Supplementary Table, please indicate the coordinates of the putative regulatory regions upstream from genes.*

1.9 We realize that our previous description of the putative regulatory regions was ambiguous and have updated it to clarify that the entire upstream and downstream region of each gene or operon was included:

In order to incorporate any possible regulatory sequences from the immediate genetic neighborhood, the entire upstream and downstream region of each gene or operon is included (upstream region: from the end of the previous gene on the genome to the beginning of the relevant gene; downstream region: from the end of the relevant gene to the beginning of the next gene on the genome).

1.10 In statement “The Multi-Drug Convolutional Neural Network (MD-CNN) comprises two convolution layers (with filter size 12 nucleotides in length), one max-pooling layer, two convolution layers, one max-pooling layer, followed by two fully-connected hidden layers each with 256 rectified linear units (ReLU) (Table 1).”. Do the authors want to refer to Figure 1 or Table 1?

1.10 Thank you for catching this typo, it is now fixed to say “Figure 1”.

1.11 *Given the increasing appreciation that drug resistance in TB is a multi-locus phenotype (as shown by recent work by this group on epistatic interactions) do the authors think their CNN models can capture these epistatic relationships systematically? Or would it be confounded by common drug resistance co-occurrences?*

1.11 This is an interesting question and was one of our main motivations implementing a deep learning model, which in theory can learn arbitrary relationships between input variables, rather than a simple linear model or catalog-based method which by definition cannot detect epistatic relationships. Here, we define epistasis as interaction between pairs or higher-order sets of mutations (ie, context dependence; a certain mutation has a different effect in combination with another mutation than it does alone). In particular, the initial filter layers of the CNN could detect local epistatic interactions, while the fully connected layers of the CNN could be learning relationships between any of the sequence motifs detected by the filters. We have not systematically analyzed the CNNs for the detection of epistasis in this work but we think it’s an exciting future possibility.

We think there are currently two main barriers to detecting epistasis systematically using a deep learning model:

1. Sufficiency of data. To successfully learn interaction parameters between any two input variants, the model would need to see many examples of those variants, both alone and in combination. There are a number of variants that confer resistance to each drug, and so detecting the effects of combinations of variants becomes a major challenge as any particular combination of variants may be seen together only rarely in the dataset. As the reviewer suggests, likely the model has detected “interactions” between the variants that confer resistance to different drugs because these combinations are common in the dataset.

2. Binarization of phenotypes may mask epistatic effects. By simply categorizing isolates as “resistant” or “sensitive”, any quantitative variation in the resistance categories is lost. Therefore, our model is theoretically limited to detecting epistatic interactions between variants that alone do not confer resistance but do confer resistance in combination. Epistatic interactions that increase the MIC (quantitative resistance measure) further above the resistance threshold would not be detected by our model in its current form.

In fact, binarization of phenotypes can also cause linear effects to appear epistatic, because the quantitative resistance values have been passed through a nonlinear function (binarization). If variant A and variant B each cause a small increase in MIC that is alone insufficient to put the isolate over the threshold MIC value for resistance, but in combination the increase is sufficient, it would appear as though A and B have an epistatic interaction, when in fact they have a linear effect on MIC that is being transformed through the non-linear binarization function.

Reviewer #2 (Remarks to the Author):

In their paper Green, Yoon and co-workers present a deep convolutional neural network that predicts the antibiotic resistance phenotypes of Mycobacterium tuberculosis isolates. The authors have trained and cross-validated data from more than 10000 M.tuberculosis strains from ReSeqTB and the WHO Supranational Reference laboratory network, whereby each isolate was phenotyped for resistance to at least one of 13 anti-TB drugs, and wherein all drugs are represented by at least 250 phenotypes isolates.

The authors find that their presented convolutional neural networks offer similar predictive accuracy to the state-of-art prediction method (published by them some time ago), while being able to identify new potential resistance mutations.

This is particularly visible in Fig. 5 where the authors depict some of newly predicted mutations in gene products of inhA and pncA, which are of uncertain significance in the WHO catalogue, that are likely involved in resistance to Isoniazid and Pyrazinamide, respectively.

These are very important outcomes of this analyses and the authors argue that these mutations are probably involved in resistance, as their positions are very close to the active sites in 3 D models, similar to known resistance mutations.

Comment 2.1 *However, it would still be very helpful for the validation of the predictions if the authors showed some confirmation of their predictions, for example by highlighting strains in global Mtb databases that show such mutations and are reportedly resistant to the respective anti-TB drugs.*

Alternatively, the authors might also take in consideration potential collaborations with wet-lab mycobacteriologists that could produce mycobacterial strains with the proposed mutations and see whether these had indeed some impact on respective anti-TB resistance characteristics, and/or try to raise resistant mutants against the respective drugs and see if the predicted mutations might be part of the mutant spectrum observed.

Such data would make the manuscript less speculative than it is at the current stage where the predictions have not been confirmed by literature or experimental data. It is true that

the later confirmation steps would need a substantial amount of time, especially for slow growing mycobacteria, but may be the authors could also find alternative, faster models to show these links.

Response 2.1: We have undertaken an experiment to validate the 18 variants identified by our model that were not previously associated with resistance. We analyzed the isolates in the newly included CRyPTIC dataset (see **Response 3.2**), which were not used during training of the model, and found that 13/18 of the variants we detect, as well as all of those highlighted in figure 5, are much more commonly found in resistant strains than sensitive strains (80-100% of isolates harboring the mutation are resistant). We highlight this in a new section in the results:

New potential predictors segregate between resistant and sensitive isolates

To further validate the 18 high-saliency positions not previously designated as resistance-conferring, we ask whether they are unevenly distributed between resistant and sensitive isolates using the held-out CRyPTIC dataset. Indeed, 13 of 18 positions have minor alleles that segregate between resistant and sensitive isolates (defined as >80% resistant isolates among isolates with the minor allele). In particular, of all occurrences of minor alleles in the previously highlighted InhA I21 and I194 positions, 100% and 91% are found in resistant strains, and of the top three unknown mutations in pncA, G78, T100, and R140, 94%, 90%, and 88% are found in resistant strains.

We have further validated our models by assessing them on an additional hold-out dataset, CRyPTIC, as well as down-sampling the CRyPTIC dataset to reflect real clinical proportions of resistant isolates (see **Response 3.2**).

Due to the time and resource constraints of performing *in vitro* experiments in mycobacteria, we feel that such studies are beyond the scope of this manuscript.

Reviewer #3 (Remarks to the Author):

This work proposes a DNN that provides interpretability. By doing so they achieve similar performance to previous NN publications but provide insights on what is learned.

The work is performed and presented state-of-the-art. Below are some comments that may be of interest to the authors.

Comment 3.1. *While it is true that classic drug resistance testing takes a long time, there is a good implementation of point of care detection of drug resistance that serves the purpose of detecting resistance to rifampin that triggers a different pattern of care. It is not here that the problem resides, and where genome sequencing and DNN are going to change practice. Rather, it is the fine dissection of complex drug resistance that may benefit, in a second step, the long term treatment decisions.*

Response 3.1: The reviewer makes a good point that point-of-care drug resistance testing is currently implemented for rifampicin, and so the utility of our method is not necessarily that it will speed turn-around time relative to those methods. We instead argue that our method is useful for three reasons: (1) it can inform and expand the catalog of known resistance-conferring

mutations, that could in the future be used in more advanced point-of-care detection and comprehensive diagnosis of resistance to multiple drugs, (2) it could be potentially implemented as its own system complementary to current point-of-care detection methods, and (3) it can help researchers better understand resistance mechanisms across the genome to inform future research. We have clarified this point in the introduction as follows:

Molecular diagnostic tests for M. tuberculosis antimicrobial resistance reduce the time to result to hours or days, but only target a small number of loci relevant to a few antibiotics, and cannot detect most rare genetic variants(3). Although whole genome sequencing-related diagnostic tests offer the promise of testing many loci and inferring resistance to any drug, statistical association techniques have seen limited success, hindered by their inability to assess newly observed variants and epistatic effects(3–7). More complex models such as deep learning provide promising flexibility but are often uninterpretable, making them difficult to audit for safety purposes (8, 9). Moreover, interrogating black box models offers the opportunity for hypothesis generation which can be later validated, potentially improving scientific understanding of the underlying phenomenon (10). An ideal sequencing-based diagnostic method would predict resistance to any drug based on the entire genome, and rapidly provide interpretable outputs about which loci contributed to resistance prediction, allowing for such a method to greatly augment current molecular diagnostics with expanded catalogs of resistance-conferring loci, or supersede those diagnostics entirely.

Comment 3.2. *The observation that there is a drug-drug correlation structure may need more text. It works in this setting because the training (and testing datasets) are enriched for resistance and multidrug resistance may be a significant obstacle to generalizing the model to the real world pattern of resistance in the general population infected with conventional circulating strains. One possible approach would be to test the models against a testing set that reflects the patterns in the general testing laboratories. Otherwise, a model trained on enriched datasets may only be fit for a reference laboratory that tests high numbers of resistant, MDR and extended MDR strains.*

Response 3.2 We have undertaken this suggestion and analyzed the performance of our models on the CRyPTIC dataset, an additional held-out dataset of phenotype and genotyped MTB strains that was recently released, and a downsampled version of the CRyPTIC dataset built to reflect realistic resistance proportions. We found the performance to hold (and even be slightly better) on the resistance down-sampled dataset relative to the whole CRyPTIC dataset. We report this in a new results section:

CNN models generalize on new data with realistic resistance proportions

Datasets of drug resistance phenotypes in M. tuberculosis are enriched in resistant isolates compared to real global resistance frequencies. To assess the generalizability of our methods in a more realistic setting, we generate down-sampled, hold-out test data from CRyPTIC with 95% pan-susceptible isolates and 5% rifampicin-resistant isolates to approximate the global prevalence (22) (Methods). CRyPTIC provides genotype and phenotype data for the first-line drugs rifampicin, isoniazid, and ethambutol; and for the second-line drugs levofloxacin, amikacin, moxifloxacin, kanamycin, and ethionamide.

On the down-sampled CRyPTIC dataset, the MD-CNN produces mean sensitivities of 97.7% (CI 97.5 – 97.9) [first-line drugs], and 86.7% (CI 85.8 – 87.7) [second-line drugs]; and mean specificities of 97.4% (CI 97.2 - 97.7) [first-line drugs], and 98.5% (CI 97.5 – 99.4) [second-line drugs]. For the SD-CNN, mean sensitivities are 96.2% (CI 95.9 – 96.5) [first-line drugs], and 87.7% (CI 86.8 – 88.5) [second-line drugs]; mean specificities are 97.5% (CI 97.2-97.8) [first-line drugs], and 98.9% (CI 98.1 - 99.7) [second-line drugs].

When applied to the whole hold-out test CRyPTIC data, the models have lower performance, particularly specificity: the MD-CNN's mean sensitivities are 96.4% (CI 96.36 – 94.44) [first-line drugs] and 83.3% (CI 83.1 – 83.4) [second-line drugs]; mean specificities are 92.7% (CI 92.66 – 92.74) [first-line drugs] and 92.6% (CI 92.5 – 92.8) [second-line drugs]. For the SD-CNN, mean sensitivities are 95.3% (CI 95.2 - 95.3) [first-line drugs] (CI and 85.4% (CI 85.3 – 85.6) [second-line drugs]; mean specificities are 94.9% (CI 94.8 – 94.9) [first-line drugs] and 96.9% (CI 96.8 – 97.0) [second-line drugs].

And interpret the findings in the discussion:

*Though neural networks are often criticized for lack of interpretability, we undertake in silico mutagenesis experiments to understand the behavior of our network. By computationally introducing resistance-conferring mutations into known susceptible sequences, we discover that the MD-CNN's predictions for second-line drugs relies on the correlation structure of drug resistance which is present in both the training and test set. Such correlations have been previously observed and shown to improve molecular diagnostic accuracy in *M. tuberculosis* (31). Indeed, these correlations appear to improve the MD-CNN's sensitivity when predicting resistance to second-line drugs.*

To investigate whether the correlation structure of drug resistance would prove a hindrance to the generalizability of the MD-CNN, we generated a dataset with realistic resistance proportions by downsampling the held-out CRyPTIC dataset. We find that both deep learning models perform well on samples with realistic resistance rates, particularly in terms of specificity. We propose that the correlation structure of drug resistance is a useful feature for achieving high sensitivity models, and is not problematic in more diverse datasets with lower resistance rates. However, should antibiotic usage guidelines change or isolates with unusual resistance patterns be sequenced, the models may require revisiting to ensure that correlation structure does not become a hindrance.

Comment 3.3 *The manuscript correctly implements concepts of circulating lineages and how much this genetic information contributes to the results. However, this is an important potential confounder and may need more text to reassure the reader that this is adequately controlled for.*

Response 3.3: We have taken the suggestion to expand the discussion of lineage-defining variants in our results section, as follows:

*We then determine whether the high saliency positions may be indicative of *M. tuberculosis* population structure using predetermined sets of 62 and 95 lineage-defining genetic variants (27, 28), respectively, and variants in perfect linkage with a lineage-defining variant (**Methods**). These variants comprise only a small proportion of the most*

salient positions, ranging from 0% to 8% of the top 0.1% of hits for each locus (**Supplementary Table 3**). We note that although the majority of lineage defining variants are not associated with resistance, some have been causally associated, including recently to the drug pretomanid (doi: [10.1093/jac/dkac070](https://doi.org/10.1093/jac/dkac070)).

Comment 3.4 *The authors may use some collapsing of drug class members to enhance the learning. For example, all fluoroquinolones can be collapsed together because they are felt to share to large extent the mechanisms of resistance.*

Response 3.4: We appreciate the suggestion to collapse drugs of the same class to increase dataset size, and it was an idea we had originally considered. However, we think this would obscure important differences, as not all strains exhibit cross-resistance to all drugs of the same class. For example, fluoroquinolones have been shown to have low concordance in diagnostic susceptibility test results - *ie*, many isolates measure as resistant to at least one but not all fluoroquinolones (<https://doi.org/10.5588/ijtld.14.0814>). Moreover, some mutations in *gyrB* have been shown to have different effects on resistance to early versus later generation fluoroquinolones (<https://doi.org/10.1128/JCM.02775-15>). Therefore, we decided to avoid collapsing drug classes because drugs of the same class are not interchangeable.

Comment 3.5 *The identification and mapping to protein structures of candidate resistance variants is a very important contribution of this study. Thus, the value of the work would be maximized if the authors tested a couple of the proposed variants in vitro. It is clear that this is not done overnight – but it would be great if this could be included in the manuscript.*

Response 3.5 We have taken a number of additional steps to validate our models and findings during revision, including assessing the performance of our models on a new held-out dataset and a downsampled version of that dataset (**Response 3.2**), and analyzing the candidate resistance variants for their presence in resistant versus sensitive isolates in a new dataset (**Response 2.1**). Both assessments validated the generalizability of our models and the potential utility of the putative resistance variants.

Due to the resource and time constraints of performing *in vitro* experiments in *Mycobacterium tuberculosis* or related model species, which can take at least 9 months, performing *in vitro* experiments would be prohibitive.

Comment 3.6 *Last point refers to clinical communication of the results. AUCs are the workhorse of this type of work. However, clinical laboratories and medical staff use the concepts of sensitivity, specificity, pos predictive value etc. It would be useful to speak to this audience presenting a reasonable translation of the performance of such tools in the clinical laboratory setting.*

Response 3.6: We agree with the suggestion and have included a new section in the manuscript discussing sensitivity and specificity of the ML models as well as the WHO variant catalog (see Response 1.2):

An important feature of the CNN models is the ability to tune the model threshold to optimize sensitivity or specificity, depending on the application. We choose a threshold for all of our machine learning models (MD-CNN, SD-CNN, logistic regression + L2,

and WDNN) that maximizes the sum of sensitivity and specificity. We find that the MD-CNN has the highest sensitivity of the four models for first-line (mean sensitivity 91.9%) and second-line drugs (mean sensitivity 91.1%) except ethambutol, for which the WDNN exhibits the highest sensitivity (**Supplementary Table 2**). The SD-CNN demonstrates the greatest specificity for first-line drugs (mean specificity 94.1%) except ethambutol where the MD-CNN has the highest; the SD-CNN demonstrates the highest specificity for second-line drugs (mean specificity 94.3%) except ethionamide and ciprofloxacin where the MD-CNN out-performs (**Supplementary Table 2**).

Reviewers' Comments:

Reviewer #1:

Remarks to the Author:

The authors have conducted a great deal of work to address this rebuttal. I am largely happy with their answers to my comments, but I still have a few minor comments I'd like them to consider, mostly about bringing conclusion statements from the rebuttal to the main manuscript.

- In statement: "The WDNN was also found to have higher sensitivity than existing catalog-based methods in a recent comparative study (21).", please indicate in brackets what these methods are (i.e. Mykrobe and TB-Profiler).

- The authors should bring this conclusion statement to the main manuscript: "We find that the catalog-based method has reduced sensitivity compared to our deep learning methods. This is not surprising as the catalog was designed to be very selective about which variants are labeled as resistance-conferring, and hence is much more conservative in calling resistance, highlighting the need for more sophisticated methods."

- The following rebuttal point should be included in the Discussion: "Our new analysis of sensitivity and specificity shows that the MD-CNN has the highest sensitivity of any model analyzed, but that the sensitivity comes at the expense of lowered specificity (compared to the SD-CNN for example). This supports the reviewer's suggestion that the MD-CNN, while highly sensitive, is more likely to call false positive resistances, possibly due to the use of the drug-drug correlation structure by the model."

- The authors' answer to question 1.11 is very insightful so it's a pity it has not been included in the main manuscript. They could mention in the Discussion the future need to predict MICs from WGS data and to discover additive and epistatic interactions predictive of drug resistance, a limitation of current approaches (including catalogue-based).

Reviewer #2:

Remarks to the Author:

The authors have responded adequately to my and other reviewer's questions and concerns.

Reviewer #3:

Remarks to the Author:

The authors have provided careful answers to the queries.

There is however, one answer to Reviewer 1 (Respose 1.5) where the authors indicate that "MD-CNN, while highly sensitive, is more likely to call false positive resistances". This is a potential issue in clinical application as it would restrict the use of appropriate drugs. There should be a comment on this.

Reviewer #1 (Remarks to the Author):

The authors have conducted a great deal of work to address this rebuttal. I am largely happy with their answers to my comments, but I still have a few minor comments I'd like them to consider, mostly about bringing conclusion statements from the rebuttal to the main manuscript.

Response: We thank the reviewer for their time and insightful comments that have greatly improved the manuscript.

Comment 1.1: In statement: "The WDNN was also found to have higher sensitivity than existing catalog-based methods in a recent comparative study (21).", please indicate in brackets what these methods are (i.e. Mykrobe and TB-Profiler).

Response 1.2: The method names have now been indicated in parentheses.

Comment 1.2: The authors should bring this conclusion statement to the main manuscript: "We find that the catalog-based method has reduced sensitivity compared to our deep learning methods. This is not surprising as the catalog was designed to be very selective about which variants are labeled as resistance-conferring, and hence is much more conservative in calling resistance, highlighting the need for more sophisticated methods."

Response 1.2: We have added an additional paragraph in the discussion to address this:

We compare the sensitivity and specificity of our deep learning methods to the current WHO catalog-based method, and find that the catalog method has lower sensitivity. This is not surprising as the catalog was designed to be very selective about which variants are labeled as resistance-conferring, and hence is much more conservative in calling resistance, highlighting the need for machine learning methods. Machine learning methods also have tunable thresholds that can be optimized for favor sensitivity or specificity depending on the application, a feature that is absent in catalog-based methods.

Comment 1.3: The following rebuttal point should be included in the Discussion: "Our new analysis of sensitivity and specificity shows that the MD-CNN has the highest sensitivity of any model analyzed, but that the sensitivity comes at the expense of lowered specificity (compared to the SD-CNN for example). This supports the reviewer's suggestion that the MD-CNN, while highly sensitive, is more likely to call false positive resistances, possibly due to the use of the drug-drug correlation structure by the model."

Response 1.3: We have added an additional paragraph in the discussion to address this point:

Our analysis of sensitivity and specificity shows that the MD-CNN has the highest sensitivity of any model analyzed, but that the sensitivity comes at the expense of lowered specificity. This suggests that the MD-CNN, while highly sensitive, is more likely to call false positive resistances, possibly due to the use of the drug-drug correlation structure by the model. In contrast, the SD-CNN has no access to the drug-drug correlation structure and demonstrates the greatest specificity. It may be useful to use both types of CNNs sequentially in clinical settings: if the MD-CNN predicts susceptibility to a drug, the SD-CNN is not consulted; if the MD-CNN predicts resistance to a drug, resistance can be confirmed or disputed through a second prediction from the SD-CNN.

Comment 1.4: The authors' answer to question 1.11 is very insightful so it's a pity it has not

been included in the main manuscript. They could mention in the Discussion the future need to predict MICs from WGS data and to discover additive and epistatic interactions predictive of drug resistance, a limitation of current approaches (including catalogue-based).

Response 1.4. We have endeavored to summarize our response in a new paragraph in the discussion section:

In addition, applying sequence filters in the convolutional layers allows for pooling of variant effects, such that any deviation from wild type sequence structure can be more directly associated with resistance even when individual variants are rare. Deep learning methods can also theoretically learn arbitrary interaction terms between input genetic variants and this may further explain performance gains over simpler models like logistic regression. Further study, perhaps using in silico mutagenesis, is required to determine which, if any, epistatic interactions are captured by these models. In this study the detection of epistasis is limited by the use of binary resistance phenotype data, which masks any epistatic effect that further increases resistance of an already resistant strain. It is hence possible that deep learning models may show further performance improvements over regression methods when applied to quantitative resistance data, where epistatic relationships between variants may be more apparent.

Reviewer #2 (Remarks to the Author):

Comment 2: The authors have responded adequately to my and other reviewer's questions and concerns.

Response 2: We thank the reviewer for their time and helpful comments during the revision process.

Reviewer #3 (Remarks to the Author):

The authors have provided careful answers to the queries.

Comment 3.1: There is however, one answer to Reviewer 1 (Response 1.5) where the authors indicate that "MD-CNN, while highly sensitive, is more likely to call false positive resistances". This is a potential issue in clinical application as it would restrict the use of appropriate drugs. There should be a comment on this.

Response 3.1: We have added an additional paragraph in the discussion to address this point:

Our analysis of sensitivity and specificity shows that the MD-CNN has the highest sensitivity of any model analyzed, but that the sensitivity comes at the expense of lowered specificity. This suggests that the MD-CNN, while highly sensitive, is more likely to call false positive resistances, possibly due to the use of the drug-drug correlation structure by the model. In contrast, the SD-CNN has no access to the drug-drug correlation structure and demonstrates the greatest specificity. It may be useful to use both types of CNNs sequentially in clinical settings: if the MD-CNN predicts susceptibility to a drug, the SD-CNN is not consulted; if the MD-CNN predicts resistance to a drug, resistance can be confirmed or disputed through a second prediction from the SD-CNN.